# Structural and biochemical insights into small RNA 3′ end trimming by *Arabidopsis* SDN1

Jiayi Chen [1], Li Liu[2], Chenjiang You [2], Jiaqi Gu[1], Wenjie Ruan[1], Lu Zhang[1], Jianhua Gan[3], Chunyang Cao[4], Ying Huang [5], Xuemei Chen[2] & Jinbiao Ma [1]

A family of DEDDh 3′→5′ exonucleases known as Small RNA Degrading Nucleases (SDNs) initiates the turnover of ARGONAUTE1 (AGO1)-bound microRNAs in *Arabidopsis* by trimming their 3′ ends. Here, we report the crystal structure of *Arabidopsis* SDN1 (residues 2-300) in complex with a 9 nucleotide single-stranded RNA substrate, revealing that the DEDDh domain forms rigid interactions with the N-terminal domain and binds 4 nucleotides from the 3′ end of the RNA via its catalytic pocket. Structural and biochemical results suggest that the SDN1 C-terminal domain adopts an RNA Recognition Motif (RRM) fold and is critical for substrate binding and enzymatic processivity of SDN1. In addition, SDN1 interacts with the AGO1 PAZ domain in an RNA-independent manner in vitro, enabling it to act on AGO1-bound microRNAs. These extensive structural and biochemical studies may shed light on a common 3′ end trimming mechanism for 3′→5′ exonucleases in the metabolism of small non-coding RNAs.

[1] State Key Laboratory of Genetic Engineering, Collaborative Innovation Centre of Genetics and Development, Department of Biochemistry, Institute of Plant Biology, School of Life Sciences, Fudan University, Shanghai 200438, China. [2] Department of Botany and Plant Sciences, Institute of Integrative Genome Biology, University of California, Riverside, CA 92521, USA. [3] State Key Laboratory of Genetic Engineering, Collaborative Innovation Center of Genetics and Development, Department of Physiology and Biophysics, School of Life Sciences, Fudan University, Shanghai 200438, China. [4] State Key Laboratory of Bio-organic and Natural Product Chemistry, Shanghai Institute of Organic Chemistry, Chinese Academy of Sciences, Shanghai 200032, China. [5] State Key Laboratory of Molecular Biology, National Center for Protein Science Shanghai, Shanghai Science Research Center, Shanghai Key Laboratory of Molecular Andrology, CAS Center for Excellence in Molecular Cell Science, Shanghai Institute of Biochemistry and Cell Biology, Chinese Academy of Sciences, Shanghai 200031, China. Correspondence and requests for materials should be addressed to Y.H. (email: huangy@sibcb.ac.cn) or to X.C. (email: xuemei.chen@ucr.edu) or to J.M. (email: majb@fudan.edu.cn)

MicroRNAs (miRNAs) are a class of endogenous small RNAs that impact nearly all biological processes by controlling gene expression at the posttranscriptional level. The critical biological functions of miRNAs necessitate tight control of their own abundance in vivo. Both biogenesis and degradation contribute to the steady-state levels of miRNAs. The biogenesis of miRNAs is a multi-step process that results in a mature miRNA loaded into its effector ARGONAUTE (AGO) protein to form the RNA-induced silencing complex (RISC)[1]. Within RISC, the 5′ and 3′ ends of the miRNA are accommodated by binding pockets in the AGO protein[2,3].

The degradation of AGO-bound small RNAs in vivo entails their 3′ trimming and tailing. In plants, miRNAs and siRNAs are methylated by the small RNA methyltransferase HEN1 to protect them from degradation[4]. In *hen1* mutants, full-length miRNAs are greatly reduced in abundance while miRNA species with 3′ truncation and 3′ uridylation are produced[5]. These species are associated with the miRNA effector AGO1[6,7], suggesting that trimming and tailing occur while the miRNAs are AGO1-bound. In *Drosophila*, endogenous siRNAs associated with Ago2 are methylated by dmHen1[8,9]. In *dmhen1* mutants, Ago2-bound siRNAs are 3′ truncated and uridylated[10]. In *Arabidopsis*, the nucleotidyl transferases HESO1 and URT1 are responsible for miRNA uridylation while the exonucleases SDN1 and SDN2 are responsible for miRNA 3′ trimming[11–15]. In vitro, SDN1 can almost completely degrade free miRNAs, generating products of a very small size[16], but can only trim AGO1-bound miRNAs by a few nucleotides[15]. This trimming activity is critical as it removes the last nucleotide that is 2′-*O*-methylated to allow HESO1 and URT1 to act on the unmethylated, trimmed species[15].

Target or target mimic RNAs induce the degradation of RISC-associated small RNAs. In *Drosophila* and *Arabidopsis*, highly complementary target or target mimic RNAs lead to the specific degradation of cognate miRNAs in vivo[10,17–20]. In fly extracts, fully complementary target RNAs cause the 3′ trimming and tailing of cognate, Ago1-bound miRNAs[10]. Structural studies reveal that a fully complementary target dislodges the 3′ end of the guide strand from the PAZ domain of an AGO protein[21]. Highly complementary target RNAs are also known to completely dislodge miRNAs from human Ago2 in vitro[22]. Despite evidence implicating the role of target RNAs in miRNA degradation, it is not known how miRNA trimming or tailing enzymes engage target RNAs or AGO proteins to exert their effects on AGO-bound miRNAs.

*Arabidopsis* SDN1 and SDN2 belong to the DEDD 3′→5′ exonuclease superfamily, characterized by an active core consisting of three separate sequence motifs with four invariant acidic amino acids (DEDD). The DEDD superfamily can be further divided into two subfamilies, DEDDh and DEDDy, distinguished by a histidine or a tyrosine present in motif III. The *Arabidopsis* SDN family belongs to the DEDDh subfamily. Many DEDD superfamily members act in the 3′ end trimming or maturation of small non-coding RNAs. For example, Eri-1 was identified in *C. elegans* for its negative effects on RNA interference[23, 24]. It trims 2–4 nucleotides from the 3′ overhang of an siRNA/siRNA* duplex in vitro, thus making the duplex ineffective in RNA silencing;[23, 24] it is also involved in 26 G endo-siRNA maturation[25]. Triman is another DEDD exonuclease that functions in the 3′ end maturation of some Dicer-independent small RNAs connected to heterochromatin formation in *S. pombe*[26]. *Neurospora crassa* QIP was found to interact with the Argonaute protein QDE-2 and trim the 3′ end of QDE-2-loaded precursors of miRNA-like RNAs (milRNAs) into mature milRNAs together with the exosome[27,28]. *Drosophila* Nibbler is a DEDD exonuclease that functions in the 3′ end maturation of AGO1-bound miRNAs[29,30] and PIWI-interacting RNAs (piRNAs)[31,32].

PARN-1 in *C. elegans* and its homolog PNLDC1 in *Bombyx mori* are DEDD exonucleases that trim piRNA precursors to generate mature piRNAs[33,34]. AGO/PIWI-mediated protection and DEDD exonuclease-mediated truncation may be universal mechanisms of small RNA 3′ end formation or small RNA degradation.

To understand the mechanistic basis of SDN1's activities in miRNA metabolism, we conducted structural and biochemical studies. Here, we report the crystal structure of *Arabidopsis* SDN1 (residues 2–300) in complex with a 9 nt single-stranded (ss) RNA at 2.8 Å resolution and the crystal structure of SDN1 C-terminal domain (CTD) at 2.05 Å resolution. The SDN1 DEDDh catalytic domain interacts extensively with the N-terminal domain (NTD) and binds four nucleotides from the 3′ end of the RNA via its catalytic pocket. The structure of the SDN1 CTD revealed a non-canonical RRM fold with two extended β strands. Through biochemical studies, we showed that the C-terminal RRM domain acts cooperatively with the DEDDh domain in substrate recognition and is critical for enzymatic processivity. We revealed that the RRM domain binds the 5′ regions of ssRNA substrates or target strands of miRNA/target RNA duplexes, in coordination with the DEDDh domain that attacks the 3′ ends of miRNAs. In addition, we showed that SDN1 interacts with the PAZ domain of AGO1 in an RNA-independent manner and the RRM domain is critical for the trimming of AGO1-bound miRNAs in vitro. Our work reveals how SDN1 acts on free miRNAs and provides a model on how it might trim AGO1-bound miRNAs.

## Results

**Structure of SDN1 in complex with an ssRNA substrate.** Full-length, recombinant *Arabidopsis* SDN1 (residues 2–409) and a 10 nt ssRNA (5′-AGCCCAUUAG-3′) were used in crystallization. The crystals diffracted up to 2.8 Å and the structure was refined to an $R_{work}$ of 25.2% and an $R_{free}$ of 26.1% with good stereochemistry (Table 1). However, only the N-terminal domain (NTD; residues 2–137) and the DEDDh catalytic domain (residues 138–300) of SDN1 (Fig. 1a), plus nine nucleotides from the 3′ end of the RNA was clearly observed in the structure (Fig. 1b and Supplementary Fig. 2a). The 5′ region of the RNA substrate interacts with another SDN1 molecule in an adjacent asymmetric unit due to crystal packing (detailed in Supplementary Fig. 2a-c and legends). The C-terminal region of SDN1 (residues 301–409) lacked observable density, but was not cleaved during crystallization (Supplementary Fig. 2d,e). Hence, the structure is referred to as SDN1 ΔC-ssRNA hereafter.

In the structure, SDN1 NTD is closely attached to the DEDDh domain through a large interface (1379.2 Å²), and the DEDDh domain interacts with the RNA substrate (530.0 Å²; Fig. 1c, d). SDN1 NTD is composed of seven α helices and could not be attributed to any characterized protein family by amino acid sequence or domain architecture. Various hydrophobic interactions and hydrogen bonds (H-bonds) were observed between the NTD and the DEDDh domains, enabling these two domains to form a rather rigid structure. (detailed in Fig. 1e-g and legends). Expression of the SDN1 DEDDh domain alone (residues 137–302) in *E. coli* resulted in extremely low yield and severe degradation, suggesting that the massive inter-domain interactions stabilize the DEDDh domain.

The SDN1 DEDDh domain consists of a core of a five-stranded twisted β sheet surrounded by six α helices (Fig. 1b). The core structure of the SDN1 DEDDh domain resembles other DEDDh family exonucleases, e.g., human ISG20 (hISG20, PDB code: 1WLJ), an interferon-induced antiviral ribonuclease[35], and *Neurospora crassa* Pan2 (PDB code: 4CZW), the catalytic subunit of Pan2-Pan3 deadenylation complex[36]. Structural superimposition and sequence alignment of the SDN1 DEDDh

## Table 1 Data collection and refinement statistics

| | SDN1-ssRNA (Native) | Se-Met SDN1-ssRNA (SAD) | Se-Met RRM (SAD) |
|---|---|---|---|
| Data collection | | | |
| Space group | P3₁21 | P3₁21 | C222₁ |
| Cell parameters | | | |
| a, b, c (Å) | 88.0, 88.0, 178.6 | 87.7, 87.7, 178.6 | 41.1, 65.6, 74.7 |
| α, β, γ (°) | 90, 90, 120 | 90, 90, 120 | 90, 90, 90 |
| Wavelength(Å) | 0.9793 | 0.9793 | 0.9793 |
| Resolution(Å) | 30.00–2.80 (2.90–2.80)ᵃ | 30.00–2.80 (2.90–2.80) | 34.85–2.05 (2.12–2.05) |
| $R_{merge}$ | 0.07 (0.45) | 0.10 (0.42) | 0.11 (0.31) |
| Average I/σ(I) | 31.8 (2.2) | 26.6 (2.1) | 29.4 (11.4) |
| Completeness (%) | 99.3 (99.1) | 99.2 (98.1) | 99.0 (98.9) |
| Redundancy | 11.3 (5.6) | 6.8 (4.1) | 13.7 (13.2) |
| Refinement | | | |
| Resolution (Å) | 29.00–2.8 | | 34.85–2.05 |
| No. reflections | 20209 | | 6539 |
| $R_{work}/R_{free}$ | 0.252/0.261 | | 0.178/0.227 |
| No. of atoms | | | |
| Protein | 2264 | | 780 |
| RNA | 190 | | Not applicable |
| Mg²⁺ | 2 | | Not applicable |
| Other molecules | 10 | | 26 |
| Water | 27 | | 76 |
| Average B factor (Å²) | | | |
| Protein | 102.1 | | 26.0 |
| RNA | 124.5 | | Not applicable |
| Mg²⁺ | 94.9 | | Not applicable |
| Other molecules | 148.5 | | 27.8 |
| Water | 80.5 | | 32.1 |
| RMS deviation | | | |
| Bond length (Å) | 0.015 | | 0.007 |
| Bond angle (°) | 1.843 | | 1.049 |
| Ramachandran plot (%) | | | |
| Most favored regions | 96.8 | | 97.0 |
| Allowed regions | 3.2 | | 3.0 |
| Outliers | 0.0 | | 0.0 |

ᵃNumbers in parentheses represent the highest resolution shell

domain and hISG20 reveals that the two proteins are highly similar not only in overall structures but also in amino acid sequences (Supplementary Fig. 1 and 2f).

**RNA 3′ end recognition by the SDN1 DEDDh catalytic pocket.** The catalytic pocket of the SDN1 DEDDh domain bound four nucleotides from the 3′ end of the ssRNA substrate (Fig. 1b). The backbone carboxyl group of M147 and the side chain of Arg185 formed H-bonds with the 2′ and 3′ hydroxyl groups of the 3′ terminal nucleotide G10 (Fig. 2a). Mutagenesis of Arg185 to alanine significantly reduced the enzyme's catalytic activity (Fig. 2f and Supplementary Fig. 2i), indicating the functional importance of Arg185. Similar interactions were observed between the corresponding Met14 and Arg53 in hISG20 and UMP in the active center (Supplementary Fig. 2g)[35]. The

structural observation of the 2′-OH group on the 3′ end ribose playing a role in enzyme-substrate interaction is consistent with in vitro enzymatic assays showing that SDN1 exhibited much lower catalytic activity towards 3′ end 2′-O-methylated RNAs than unmethylated ones (Supplementary Fig. 2h)[16]. In addition, two loops from the catalytic pocket were engaged in the RNA recognition. His223 and Ser224 on loop 1 (residues 223–225), which connects β4 and α3, formed H-bonds with the phosphate group of A9 and the 2′-OH group on the ribose of U8 (Fig. 2b). Loop 2 (residues 247–259) connecting α4 and α5 was partially disordered (residues 247–252 undetermined), while Arg256, Pro257, Ser258, and Leu259 were observed to form H-bonds with the ribose 2′-OH groups of U7 and U8, as well as the phosphate group of A9 (Fig. 2b, c and Supplementary Fig. 2j). Meanwhile, the carbonyl groups on the base of U8 was recognized by the sidechain of Arg256 (Fig. 2c). The detailed interactions between 3′ end of the RNA and the DEDDh catalytic pocket are summarized in Fig. 2e.

In the active center, SDN1 adopted a partially active form, in which two magnesium (Mg) ions were observed at occupancies of 0.57 for MgA and 0.97 for MgB (Fig. 2d). The side chains of the four invariant residues, Asp144, Glu146, Asp228, and Asp283, coordinated the two Mg ions to bind the scissile phosphate. However, only 35% of the side chain of Glu146 is in the active conformation that coordinates MgA, while 65% is in an inactive state that flips away from the active site (Fig. 2d). Superimposition of the active sites of SDN1 and hISG20 shows high resemblance (Supplementary Fig. 1g). Mutation of any one of the five conserved residues (Asp144, Glu146, Asp228, Asp283, and His278; Supplementary Fig. 1a and 2i) abolished the exonuclease activity of SDN1 (Fig. 2f), suggesting the functional importance of the five catalytic residues[37].

**SDN1 CTD adopts an RRM fold and enhances substrate binding.** In vitro enzymatic assays showed that deletion of the SDN1 CTD (SDN1 ΔC, residues 2–302) abolished enzymatic activity under enzyme-limiting conditions (Fig. 2f, SDN1 ΔC lane), indicating that SDN1 CTD is required for activity. Electrophoretic mobility shift assays (EMSA) showed that SDN1 ΔC was greatly compromised in RNA binding (Fig. 3a–d), indicating that the CTD participates in substrate binding. To better understand its structure and function, we conducted crystallographic studies and obtained crystals of the SDN1 CTD (residues 309–409) with mutations in three glutamic acid residues (E329A/E330A/E332A). The crystal diffracts to 2.05 Å and the structure was refined to an $R_{work}$ of 17.5% and an $R_{free}$ of 23.3% with good stereochemistry (Table 1).

In the CTD structure, residues 315–409 adopts a βαββαββ RNA Recognition Motif (RRM) fold (Fig. 3e), but the N-terminal first α helix (residues 309–314) belongs to the linker connecting the DEDDh and RRM domains. In the SDN1 RRM structure, the six-stranded, antiparallel β sheet forms a positively charged surface and packs against two α-helices (Fig. 3f). Sequence similarities between SDN1 RRM and other RRM proteins are very low, but two aromatic amino acids that are usually crucial for RNA binding in RRM family members are present in SDN1 RRM (Phe318 in β1 and Phe356 in β3; Supplementary Fig. 3a). RRM1 of human Poly(U)-binding-splicing factor 60 (hPUF60, PDB code: 5KVY) is one of the top structural homologs according to Dali server. Superimposition of the two structures shows that β1, β2, and β3 of SDN1 RRM are well superimposed to the corresponding β strands in hPUF60 RRM1 (Supplementary Fig. 3b), and so are the two conserved aromatic residues in SDN1 RRM (Phe318 and Phe356; Supplementary Fig. 3c). Nevertheless, the lengths and orientations of β4, β5, and β6 in SDN1 RRM

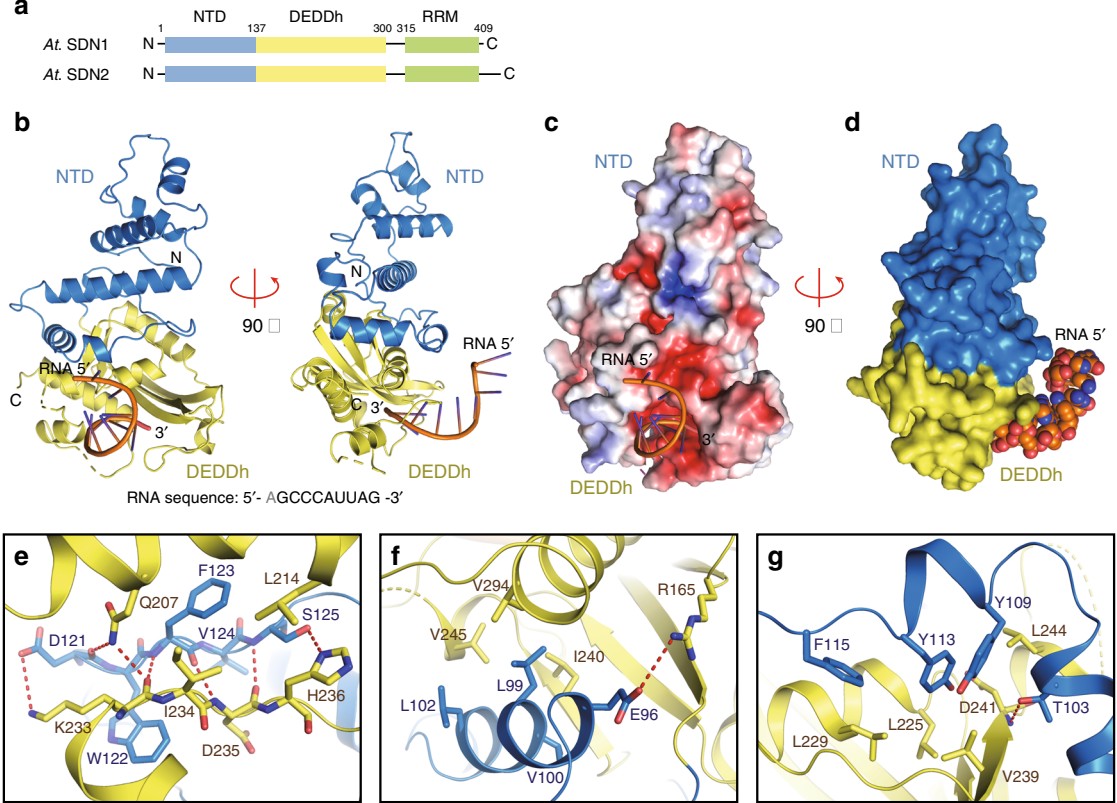

**Fig. 1** Structure of SDN1 ΔC–ssRNA complex. **a** Domain architectures of Arabidopsis SDN1 and SDN2. NTD, N-terminal domain; DEDDh, the catalytic domain; RRM, the C-terminal domain found to adopt an RRM fold in this study. The numbers of amino acid residues are as indicated. **b** Front and side views of SDN1 bound to the single-stranded RNA in cartoon mode. Domains are colored as in **a**. **c** The surface electrostatic potential of SDN1 ΔC showing that the 3′ end of the RNA is inserted into the negatively charged catalytic pocket. Red, −5.0 $k_BT/e$; Blue, + 5.0 $k_BT/e$. $k_B$, Boltzmann constant; $T$, temperature in Kalvin; $e$, charge of an electron. **d** The molecular surface of SDN1 ΔC showing the extensive interface between the NTD and the DEDDh domain. The structure also shows that the interaction of SDN1 with the RNA occurs mainly through the DEDDh domain. **e–g** Hydrophobic interactions and hydrogen bond networks between the NTD and DEDDh domains. Glu96, Thr103, Asp121, Phe123, and Ser125 in NTD form hydrogen bonds with Arg165, Gln207, Lys233, Asp235, His236, and Asp241 in the DEDDh domain (H-bonds shown in red dotted lines). The various hydrophobic interactions are depicted in **e** to **g**. **e** Phe123 interacts with Ile234 and Leu214, and the side chain of Lys233 packs against the aromatic ring of Trp122. **f** Leu99, Val100 and Leu102 on helix α5 of the NTD are buried in the hydrophobic surface formed by Ile240, Val245 and Val294 in the DEDDh domain. **g** the aromatic rings of Tyr109, Tyr113, and Phe115 protrude into the hydrophobic surface formed by Leu225, Leu229, Val239, and Leu244 in the DEDDh domain

show large differences from the corresponding tiny β strands in hPUF60 RRM1. (Supplementary Fig. 3b).

To investigate how SDN1 RRM interacts with RNA, we performed an NMR titration assay using [15]N-labeled SDN1 RRM (residues 305–409) and a 10 nt ssRNA (5′-AGCCCAUUAG-3′). During the NMR titration, various residues showed large perturbations in chemical shifts or disappearances in the cross peaks (Supplementary Fig. 3d,g). Mapping all these residues onto the RRM surface reveals that they are mainly located on the antiparallel β sheet (β1, β3, β4, β5, and β6; Fig. 3h), which is positively charged (Fig. 3f). Interestingly, NMR titration showed that Phe318 on β1 of SDN1 RRM interacts with RNA, but no large change in chemical shift was observed for Phe356, suggesting that Phe356 may not be involved in RNA binding. Another feature of SDN1 RRM is that its extended β4 and β5 participate in RNA binding (residues Thr378, Asp379, Gly382, and Gln385 detected via NMR titration; Fig. 3g), whereas most canonical RRMs consist of very short or no β4 and β5 (Supplementary Fig. 3a,b).

**SDN1 CTD enhances enzymatic processivity on ssRNA substrates.** We previously showed that SDN1 can act on 21–27 nt ssRNAs[16]. To investigate the preferred size of RNA substrates,

EMSA was performed using 5′ end [32]P-labeled RNAs of 22 nt, 16 nt, 10 nt, or 6 nt in length (Supplementary Table 1). SDN1 was found to exhibit stronger binding affinity towards small RNAs longer than 16 nt (Fig. 4a, b); the calculated apparent dissociation constant $K_{d, app}$ from EMSA assays (Fig. 3a–d) showed that neither SDN1 ΔC nor SDN1 CTD (residues 298–309) alone bound RNA well, suggesting that the DEDDh domain and the CTD may act cooperatively during substrate recognition. As the RNA 3′ end is bound by the DEDDh catalytic pocket, we hypothesize that the 5′ region of the RNA is bound by the CTD.

To test this hypothesis, we employed a UV-crosslinking assay using full-length SDN1 and SDN1 ΔC. The RNA substrate utilized in this assay was substituted at different positions with 5-iodo modified uridine (5-iodoU) (Fig. 4c), which can be crosslinked to certain amino acids (YFHM) within a distance of 3.8 Å in protein-RNA complexes upon irradiation by UV light at 312 nm[38]. This analysis allows determination of the site of the RNA that the protein contacts within an RNA-protein complex. SDN1 ΔC could be crosslinked to the 3′-IUU RNA but not the 5′-IUU RNA (Fig. 4d), consistent with the structural observation that the 3′ end of the RNA fits into the catalytic pocket. In contrast, full-length SDN1 crosslinked to not only 3′-IUU RNA, but also M-IUU and 5′-IUU RNAs (Fig. 4d). These results

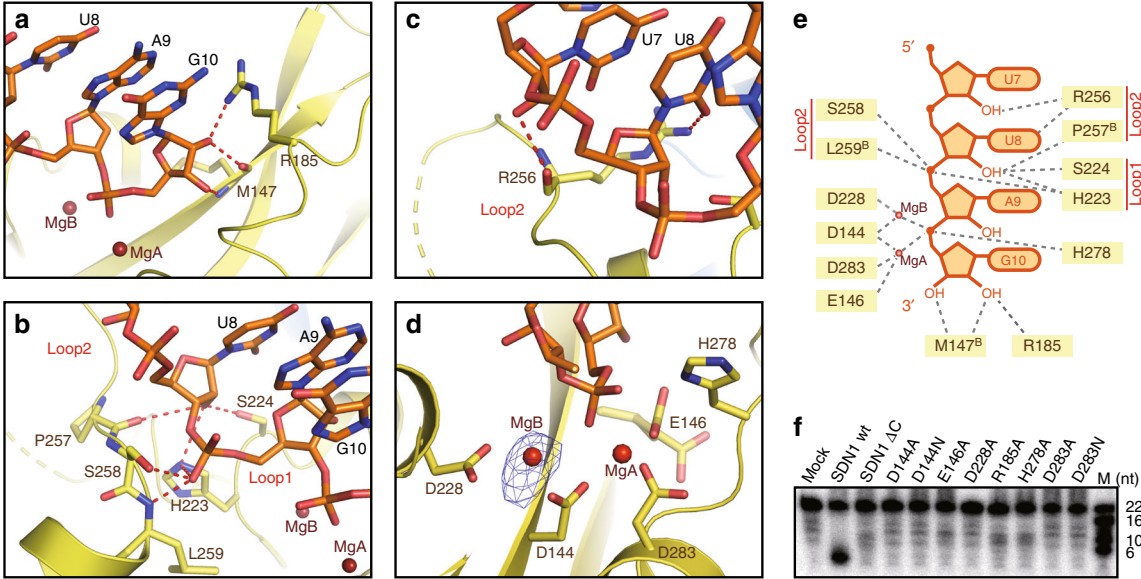

**Fig. 2** The SDN1 catalytic pocket and mechanisms of its RNA recognition. **a–c** Interactions between the SDN1 catalytic pocket and the four nucleotides from the 3′ end of the RNA. Hydrogen bonds are colored in red. **d** Active center of SDN1. Two magnesium ions are shown with the simulated annealing Fo-Fc omit map (contoured at 5.0 σ level), and there is no electron density for MgA due to low occupancy. **e** Schematic drawing of the interactions between the SDN1 catalytic pocket and the RNA substrate. **f** In vitro enzymatic activity assays with wild-type SDN1, SDN1 ΔC, and various SDN1 mutants. Wild-type SDN1 can degrade the free miRNA into products of a very small size. SDN1 ΔC and the mutants are defective in enzymatic activity

suggest that the 5-iodoUs in the middle or 5′ region of the RNAs interacted with the RRM domain, the part of protein present only in the full-length SDN1 but not in SDN1 ΔC (Fig. 4e). The crosslinking results, along with those from enzymatic (Fig. 2f) and binding assays (Fig. 3a–d), support the following model: the RNA 3′ end is seized by the DEDDh domain for hydrolysis while the RNA 5′ region is bound by the CTD to enhance substrate binding and processing, as illustrated in Fig. 4e.

According to this model, a prediction is that RRM may enhance the processivity of SDN1 on ssRNA. To test this model, we examined the processivity of wild-type SDN1 and an SDN1 mutant with the two conserved Phe residues (Phe318 and Phe356) in the RRM domain replaced by Ala (hereafter referred to as SDN1$^{2FA}$). Although Phe356 may not interact with RNA according to NMR titration and EMSA assays (Supplementary Figs. 3d and 4a,b), both SDN1$^{2FA}$ and SDN1$^{F318A}$ mutants exhibited much lower affinity and catalytic activity for miRNAs than the wild-type protein (Supplementary Fig. 4a–d) and thus could be regarded as an RRM mutant. To conduct the processivity assay, labeled ssRNA substrate was first incubated with the enzyme to allow binding. Next, Mg$^{2+}$ was added to start the reactions, with or without an excess of cold RNA added at the same time, and products of the reactions were monitored in a time course. For the wild-type enzyme, the RNA was quickly (by 1 min) degraded to a small size in the absence of cold RNA (panel 1, Fig. 4f). Even with an excess of cold competitor RNA, the wild-type enzyme generated small-sized products by 1 min and a lot of such products by 5 min (panel 2, Fig. 4f), suggesting that the enzyme was partially processive. SDN1$^{2FA}$ was much slower than wild-type SDN1 in producing small-sized products in the absence of cold RNA (panel 3, Fig. 4f), and was only able to trim 1–2 nt off the labeled RNA in the presence of cold RNA (panel 4, Fig. 4f), indicating that SDN1$^{2FA}$ dissociated from the labeled RNAs after trimming 1–2 nt off. On the other hand, the activities of wild-type SDN1 towards the labeled RNA substrates were not affected to a large degree by the cold competitor, suggesting that

the RRM domain is critical for the processivity of SDN1 on ssRNA substrates by enhancing substrate binding.

**SDN1 CTD and DEDDh cooperatively act on dsRNA substrates**. For human miRNA-Ago2 RISCs, the recognition of highly complementary target mRNAs in vitro results in the release of the miRNAs from Ago2 in the form of miRNA/target RNA duplexes[22,39]. In plants, the complementarity between miRNA and target RNAs is high, with mismatches usually at the miRNA 3′ ends[40]. Thus, it is possible that the release of miRNA/target mRNA duplexes from AGO1 occurs in plants. We performed a series of in vitro enzymatic assays using double-stranded RNAs (dsRNAs) that mimic miRNA/target mRNA duplexes to test whether SDN1 could act on such RNAs. DsRNAs with a differing number of single-stranded nucleotides at the 3′ end of the miRNA, namely miR166:T0, miR166:T2, miR166:T4, and miR166:T6, were used as substrates (Fig. 5a). SDN1 was capable of trimming several nucleotides off the miRNA strand in the miR166:T4 and miR166:T6 duplexes, forming products with 2–3 nt overhangs at the 3′ end (Fig. 5b). This finding also corroborates our structural observation that the catalytic pocket of SDN1 is 4 nt in depth, such that RNAs with shorter 3′ overhangs could not be efficiently bound (Supplementary Fig. 5c) and trimmed by SDN1.

Considering that mature miRNAs in *Arabidopsis* are 3′ end 2′-O-methylated, we additionally conducted a series of enzymatic assays using methylated miR166, carried out in the presence of either Mg$^{2+}$ or Mn$^{2+}$. Compared to assays with unmethylated miR166 RNA duplexes, the 2′-O-methyl group slowed down the reaction in the presence of Mg$^{2+}$, but SDN1 still trimmed 1 to 4 nucleotides off the miRNA strand of miR166me:T6 (Supplementary Fig. 5a). Intriguingly, in the presence of manganese, SDN1 fully degraded the miRNA strand of miR166me:T4 and miR166me:T6 regardless of the double-stranded region, while displaying very weak hydrolysis activity on miR166me:T0 and miR166me:T2 RNAs (Supplementary Fig. 5b). It seems that once

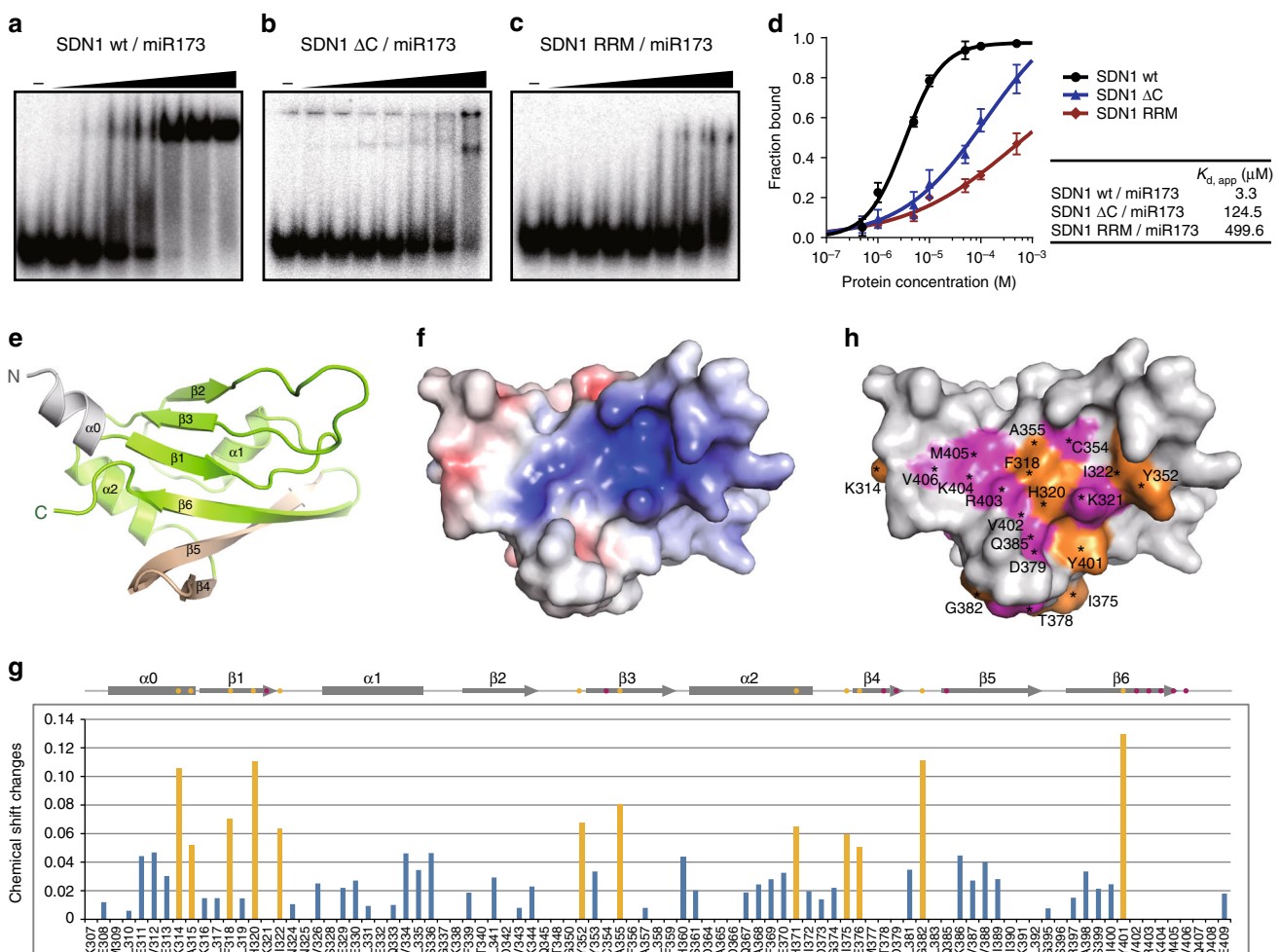

**Fig. 3** Structure and RNA binding properties of the SDN1 CTD RRM domain. **a**–**c** Electrophoretic mobility shift assays (EMSA) of wild-type SDN1 (**a**), SDN1 ΔC (**b**) and SDN1 RRM (**c**) with $^{32}$P-labeled miR173. Assays were conducted at a series of protein concentrations: 0.5 μM, 1 μM, 5 μM, 10 μM, 50 μM, 100 μM, and 500 μM (from left to right). No Mg$^{2+}$ was included to prevent catalytic activity. Scale bars represent standard deviations calculated from three biological replicates. **d** Plots showing the fraction of the protein-bound RNA at varying protein concentrations. The fraction of bound RNA was calculated from the densities of the free RNA bands in **a**, **b** and **c**. **e** Crystal structure of the SDN1 CTD E329A/E330A/E332A mutant. The N-terminal, first α helix connecting the RRM domain is colored in gray; the extended β4 and β5 in the RRM domain are shown in yellow. **f** Surface electrostatic potential of the SDN1 RRM domain showing that the antiparallel β sheet forms a positively charged surface. Red, −5.0 $k_BT/e$; Blue, + 5.0 $k_BT/e$. **g** Changes in chemical shifts for cross-peaks in the RRM spectra during NMR titration. Residues that show large chemical shift perturbations ($Δδ_{avg} > 0.05$ ppm) are colored in orange and indicated in orange dots in the diagram above. Residues of which cross peaks disappeared are indicated in pink dots in the diagram above. **h** The molecular surface of SDN1 RRM, onto which the above residues were mapped. This reveals that the antiparallel β sheet surface is responsible for RNA substrate binding. Residues that show large chemical shift changes or for which signals disappeared during titration are colored in orange or pink, respectively

the 3′ end of the RNA has access to the SDN1 active center, SDN1 exhibits extremely high hydrolytic activity in the presence of manganese.

The strong activity that SDN1 exhibits towards the miR166:T6 RNA duplex raises a question: as the 3′ single-stranded overhang is not supposed to be long enough for the simultaneous binding by both the DEDDh and the RRM domains, is RRM required for substrate binding or enzymatic activity and if so, where in the RNA duplex does RRM bind? The RRM mutant SDN1$^{2FA}$ showed greatly reduced affinity for miR166:T6 (Supplementary Fig. 5c,d), and was unable to trim off as many nucleotides as the wild-type protein (Fig. 5b, SDN1$^{2FA}$ lanes), confirming that the RRM domain is involved in RNA duplex processing. To investigate how SDN1 acts on such RNA duplexes, UV-crosslinking assays were carried out with full-length SDN1 or

SDN1 ΔC proteins and miR166:T6 RNA duplexes. The RNA duplexes were 5-iodoU modified at different positions in the complementary strand, as illustrated in Fig. 5c. Neither SDN1 nor SDN1 ΔC crosslinked to miR166:M-IUU, suggesting that SDN1 does not bind the dsRNA region. SDN1 ΔC was crosslinked to miR166:3′-IUU and very weakly to miR166:5′-IUU (Fig. 5d); the binding to 3′-IUU RNA was likely due to the interaction between the DEDDh domain and the RNA 3′ end. Full-length SDN1 was crosslinked to both miR166:3′-IUU and miR166:5′-IUU RNAs. A plausible explanation is that the DEDDh domain recognizes the miRNA 3′ overhang while the RRM domain binds the 5′ region of the complementary strand (Fig. 5e).

To verify our speculation, we conducted another UV-crosslinking assay to map the RNA-contacting domain in SDN1[38]. The recombinant SDN1 protein was redesigned with a

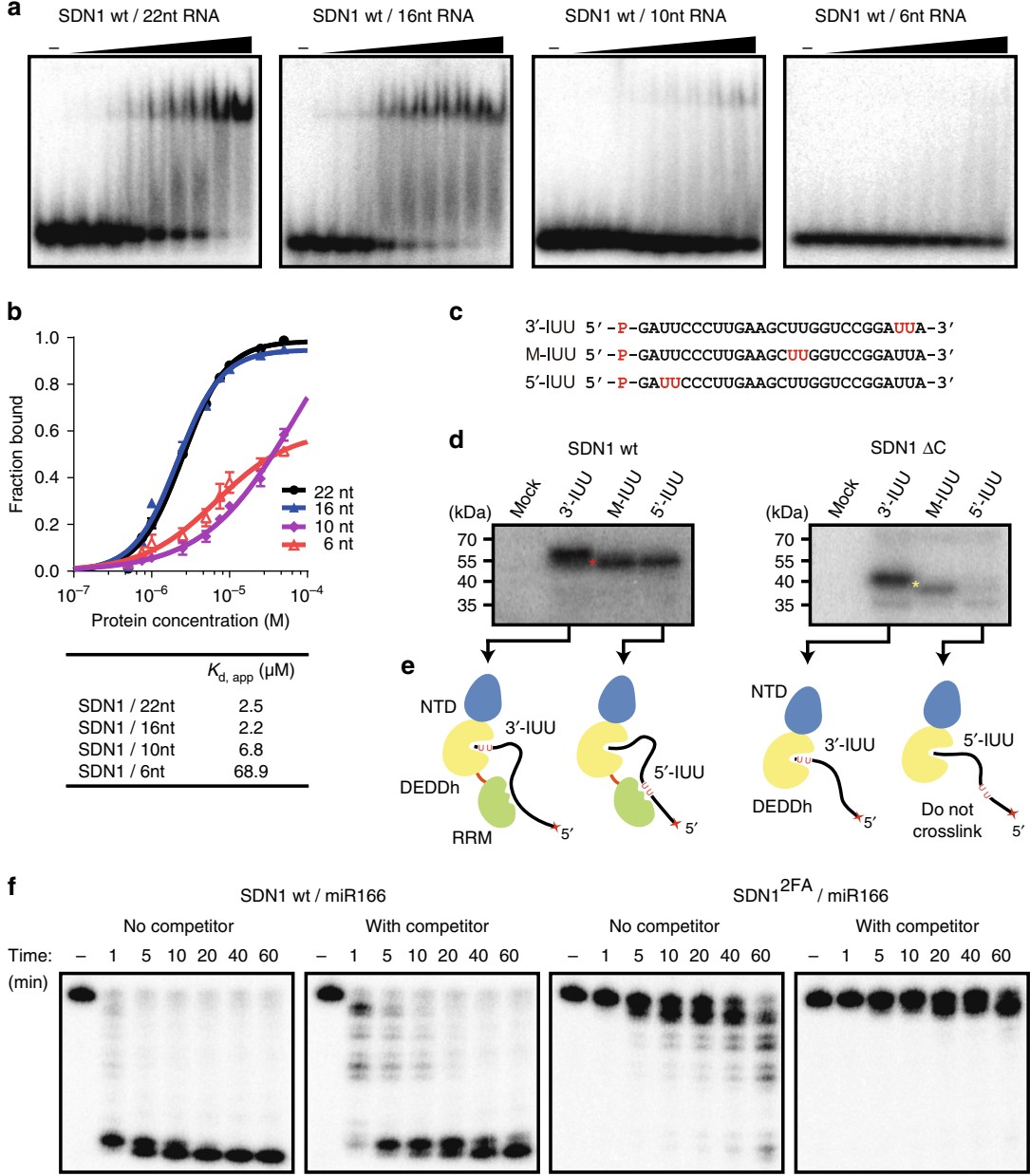

**Fig. 4** The CTD RRM domain is critical for SDN1 binding and enzymatic activities on ssRNA. **a** EMSA assays of wild-type SDN1 with 22 nt, 16 nt, 10 nt or 6 nt ssRNA substrates. Assays were conducted at a series of protein concentrations: 0.5 µM, 0.75 µM, 1 µM, 2.5 µM, 5 µM, 7.5 µM, 10 µM, 25 µM, and 50 µM (from left to right). No $Mg^{2+}$ was included to prevent catalytic activity. **b** Plots showing the RNA fractions bound at varying protein concentrations. The fractions bound were calculated from the densities of the free RNA bands in **a**. Scale bars represent standard deviations calculated from three biological replicates. **c** Diagram of 5-iodoU-substituted ssRNAs used in SDN1 UV crosslinking assays. 5-iodoUs are shown in red in the sequences. **d** UV crosslinking assays of wild-type SDN1 or SDN1 ΔC with ssRNAs shown in **c**. The red and yellow asterisks indicate full-length SDN1 crosslinked to RNA and SDN1 ΔC crosslinked to RNA, respectively. **e** Diagrams of how SDN1 or SDN1 ΔC was crosslinked to different 5-iodoU ssRNAs. **f** Enzymatic assays of wild-type SDN1 or $SDN1^{2FA}$ with or without cold RNA competitor. The labeled RNAs were first incubated with the enzymes to allow for binding but not catalysis, then $Mg^{2+}$ or a mixture of $Mg^{2+}$ and cold RNAs was added to initiate the reactions. The time indicated above each lane represents the elapsed time from the time $Mg^{2+}$ was added

TEV cleavage site inserted between the DEDDh and the RRM domains (residues: 300-ENLYFQ/S-301; Fig. 5f), and TEV cleavage was performed after UV crosslinking, so that the domain crosslinked to the RNA could be determined on an SDS-PAGE gel based on the size of the crosslinked protein-RNA complex (Fig. 5e). For the miR166:5′-IUU RNA duplex, the band after digestion matched the size of the RRM domain plus RNA, while for the miR166:3′-IUU RNA duplex, the band after digestion matched the size of the DEDDh domain plus RNA

(Fig. 5g). These results strongly support our model of how SDN1 may recognize a miRNA/target mRNA duplex, in which the SDN1 DEDDh domain binds the 3′ end of the miRNA while the RRM domain binds the 5′ region of the target mRNA (Fig. 5e).

**SDN1 binds AGO PAZ and trims the 3′ ends of AGO-bound miRNAs.** Mature miRNAs are mostly associated with AGO1 in vivo and can undergo 3′ trimming by SDN1 in vitro while

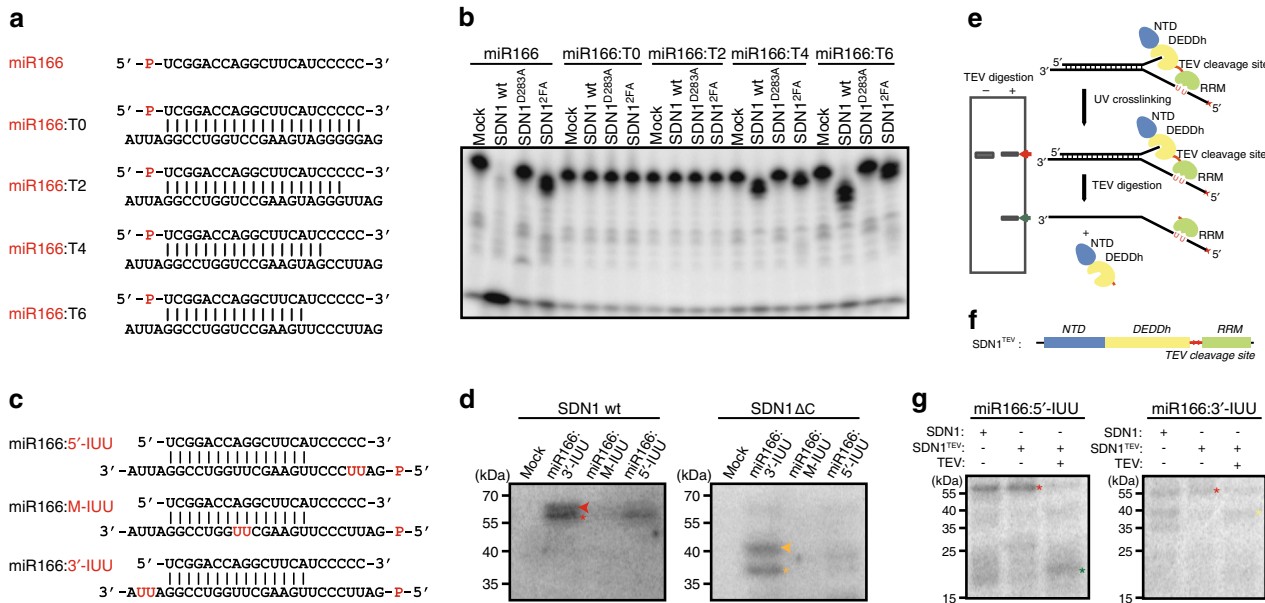

**Fig. 5** The SDN1 CTD binds 5′ region of target strands in cooperation with 3′ end recognition of miRNAs by the DEDDh domain. **a** Diagram of various dsRNAs used as substrates in SDN1 enzymatic assays. The miR166 strands were 5′ end $^{32}$P-labeled. **b** In vitro enzymatic assays with wild-type SDN1, the catalytic mutant SDN1$^{D283A}$ and the RRM mutant SDN1$^{2FA}$ on RNA duplexes. All reactions were carried out in the presence of Mg$^{2+}$. After the reactions, RNAs were denatured and resolved on denaturing polyacrylamide gels to detect the miR166 strand. **c** Diagram of various dsRNAs used in UV crosslinking assays. MiR166 is shown as the upper strand and all the target RNAs were 5′ $^{32}$P-labeled. Target RNAs were 5-iodoU modified at different positions as indicated in red. **d** UV crosslinking assays of wild-type SDN1 or SDN1 ΔC with the dsRNAs shown in **c**. The proteins were resolved on SDS-PAGE gels and visualized by autoradiography (from the $^{32}$P-label). Bands representing full-length SDN1 crosslinked to dsRNA (left panel) and SDN1 ΔC crosslinked to dsRNA (right panel) are indicated by red and orange arrowheads, respectively. The lower bands indicated by asterisks were likely crosslinking products of SDN1 proteins and ssRNAs that denatured from dsRNAs during electrophoresis. **e** Schematic diagram of the UV crosslinking assay to map the RNA-contacting domain of SDN1. SDN1 proteins were first UV crosslinked to RNAs, followed by TEV digestion and SDS-PAGE electrophoresis. The RNA-contacting domain of SDN1 is deduced by the size of the protein fragment marked by the $^{32}$P-labeled RNA. **f** Diagram of the SDN1$^{TEV}$ protein. The TEV cleavage site was inserted between the DEDDh and RRM domains. **g** UV crosslinking of SDN1$^{TEV}$ and dsRNAs followed by TEV digestion. The dsRNAs used are as indicated above the gels. Crosslinking of SDN1 or SDN1$^{TEV}$ with the same RNAs but without TEV digestion was included to indicate the size of the full-length SDN1-RNA adduct (indicated by red asterisks). Cleaved SDN1$^{TEV}$ bands at the size of the RRM domain (5-IUU RNA duplex), or at the size of SDN1 ΔC (3-IUU RNA duplex) are indicated by dark green and yellow asterisks, respectively

bound by AGO1 or AGO10[15]. We thereby asked the question of how SDN1 acts on AGO-bound miRNAs. We first examined whether SDN1 interacts with AGO1 or AGO10 through in vitro pull-down assays with recombinant, non-tagged SDN1 and MBP-tagged AGO1/10 PAZ or MID+PIWI domains (Supplementary Fig. 6a). MBP-AGO1/10 PAZ but not MBP alone or MBP-AGO1/10 MID+PIWI was able to pull down SDN1 in vitro (Fig. 6a), suggesting that SDN1 interacts with the AGO1/10 PAZ domain. The interactions between SDN1 and AGO1/10 PAZ were not disrupted by RNase A digestion (Fig. 6b), indicating that SDN1 interacts with AGO1/10 PAZ in an RNA-independent manner.

We previously showed that SDN1 can trim AGO1/10-bound miRNAs in vitro[15]. AGO1 immunoprecipitation (IP) was performed to pulldown AGO1 and associated endogenous miRNAs, and the AGO1 immunoprecipitate was used as the substrate for in vitro SDN1 enzymatic assays. Small RNA sequencing of the enzymatic products showed that SDN1 acted on many AGO1-bound miRNAs[15]. Using a similar approach, we sought to determine whether the RRM domain is required for the trimming of AGO1-bound miRNAs. First, we purified recombinant wild-type SDN1, the catalytic mutant SDN1$^{D283A}$ and the RRM mutant SDN1$^{2FA}$ (Supplementary Fig. 6b) and assessed their activities on free miRNA (Supplementary Fig. 6c). Next, AGO1 IP was performed from wild-type plants (Supplementary Fig. 6d). The SDN1 proteins were each incubated with AGO1 immunoprecipitate followed by small RNA-seq. For any miRNA species, the ratio of 3′ truncated species to full-length species was

used as a measure of the trimming activities of these enzymes. As expected, most miRNAs showed SDN1-dependent 3′ trimming, and wild-type SDN1 showed higher trimming activity than the catalytic mutant (Fig. 6d, e)[15]. Note that a few miRNAs appeared to undergo 3′ trimming in the SDN1$^{D283A}$ reactions (Fig. 6e, f). We found that the trimming was by a single nucleotide at the 3′ end (Supplementary Fig. 6f). As SDN1$^{D283A}$ is catalytically inactive, we suspect that the single nucleotide trimming was due to a contaminating nuclease in the recombinant protein preparation. In the case of wild-type SDN1, trimming occurs by several nucleotides (Fig. 6g), and this activity was abolished in SDN1$^{D283A}$ (Supplementary Fig. 6g). Although SDN1$^{2FA}$ also had some trimming activity, both the number of miRNAs undergoing trimming and the degree of trimming were reduced in the reactions with SDN1$^{2FA}$ relative to wild-type SDN1 (compare Fig. 6f to 6e). Therefore, the RRM domain is critical for the 3′ trimming of AGO1-bound miRNAs.

## Discussion

The degradation of AGO-bound small RNAs in plants and metazoans entails 3′ end truncation by exonucleases, uridylation by nucleotidyl transferases or both. The lack of structural information on 3′→5′ exonucleases that act on AGO-bound small RNAs has impeded the understanding of how these small RNAs are turned over. In this study, we solved the structure of SDN1, an exonuclease that initiates miRNA degradation by trimming their

3′ ends to generate unmethylated miRNAs that can be further turned over. Our structural and biochemical studies revealed the structural features of SDN1 in substrate recognition and provided insights on how SDN1 uses its DEDDh catalytic domain and the RRM domain cooperatively to degrade free small RNAs and trim

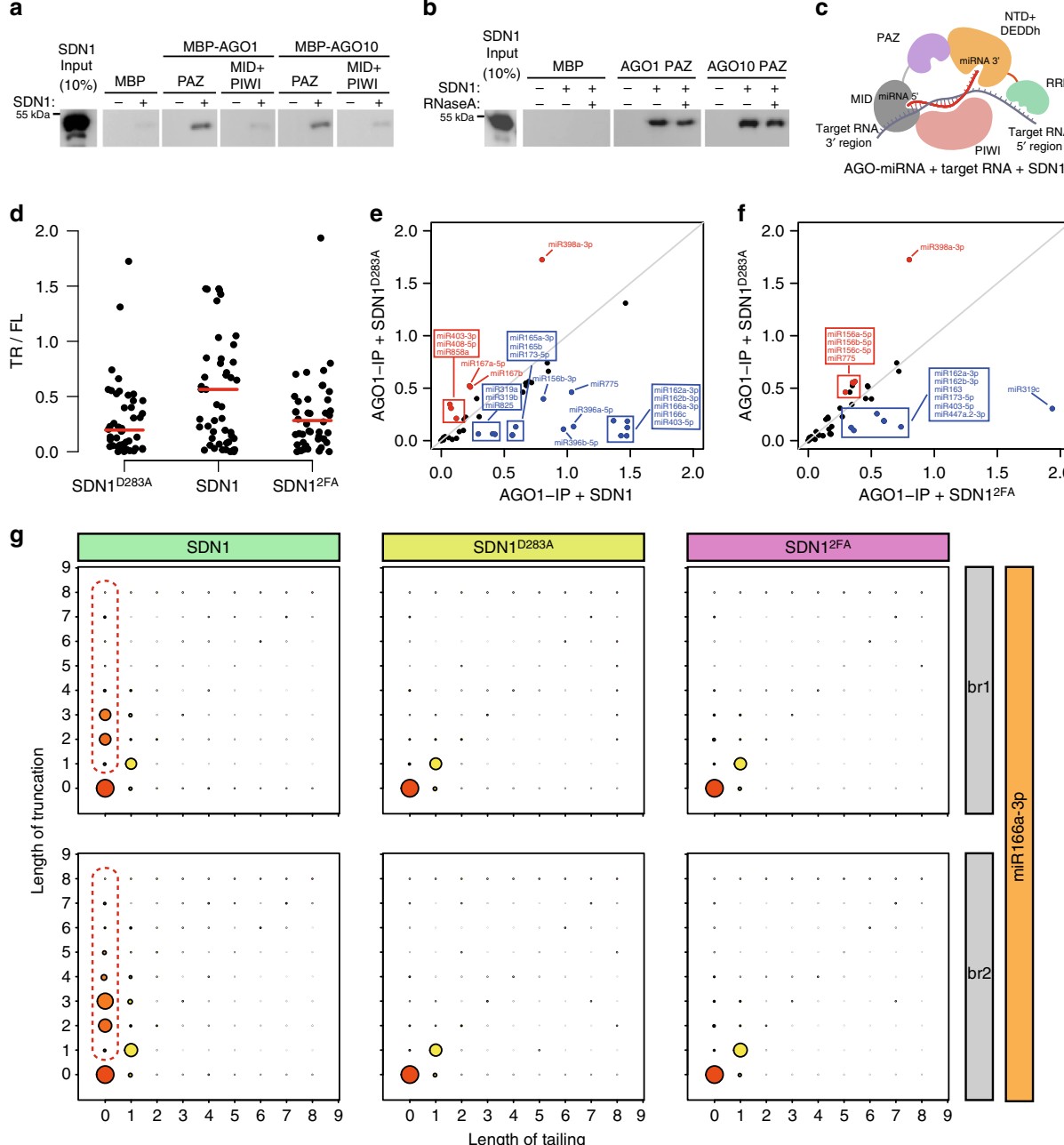

**Fig. 6** SDN1 binds the AGO1 PAZ domain and triggers the 3′ end trimming of AGO1-bound miRNAs. **a** In vitro pull-down assays to determine the interaction between SDN1 and AGO1. Recombinant SDN1 was first purified as His-SUMO-SDN1 followed by the removal of the His-SUMO tag. SDN1 was incubated with recombinant MBP-tagged *Arabidopsis* AGO1/10 PAZ or MID+PIWI on beads. Western blotting using anti-SDN1 antibodies was performed to detect SDN1 retained on beads. **b** In vitro pull-down assays as in **a** except that RNase A treatment was included to determine whether the SDN1-AGO PAZ interaction is RNA-dependent. **c** A model of SDN1 trimming AGO1-bound miRNAs in the presence of a target RNA. The miRNA 3′ end is seized by the SDN1 DEDDh catalytic pocket with the C-terminal RRM domain binding the target strand; and SDN1 interacts with the AGO1 PAZ domain to recruit SDN1 to AGO1 RISC or to facilitate the trimming process. **d–g** In vitro enzymatic assays on AGO1-bound miRNAs, followed by sRNA-seq to examine the 3′ end trimming of miRNAs. **d** Scatter plots showing the ratio of the 3′ end truncated species (TR) to the full-length species (FL) for each miRNA detected. SDN1 caused higher levels of miRNA 3′ truncation than SDN1^D283A or SDN1^2FA. **e**, **f** Plots showing the ratio of TR species to FL species for various miRNAs. The diagonal lines represent equal levels of 3′ truncated species in the SDN1^D283A and SDN1 reactions (**e**), or in the SDN1^D283A and SDN1^2FA reactions (**f**). The miRNAs marked in blue underwent SDN1-mediated 3′ truncation (**f**). **g** Diagrams showing the status of 3′ end truncation and tailing of miR166a-3p in two biological replicates (br1 and br2). The y-axis represents the number of nucleotides truncated from the 3′ end. The x-axis represents the number of nucleotides added to the 3′ end. The size of the circles indicates the relative proportions of the species. The truncated forms of miR166a-3p generated by wild-type SDN1 are highlighted in red dashed lines

miRNAs in miRNA/target RNA duplexes or miRNAs bound by AGO1.

Our structural studies uncovered a novel, seven-α helical bundle that the SDN1 NTD folds into. SDN1 NTD stabilizes the DEDDh catalytic domain and may even play a regulatory role in vivo. Other members of the SDN family also contain an NTD (Fig. 1a), thus the NTD-DEDDh interactions may be applicable to other members. The SDN1 catalytic pocket accommodates four nucleotides, which is consistent with a product size of approximately four nucleotides from longer single-stranded RNA substrates. Moreover, the size of the catalytic cavity also determines what RNA duplexes can serve as SDN1 substrates, i.e., only those with the 3′ overhangs that are long enough to reach the active site can be hydrolyzed.

The structural properties of RNA substrate binding by the SDN1 DEDDh catalytic pocket shows some common features comparing to certain DEDD homologs, e.g., the recognition of the 2′ and 3′ hydroxyl groups on the 3′ terminal ribose, which provides the structural basis for 3′ end 2′-O-methylated RNAs being suboptimal substrates for SDN1. Other enzyme-substrate interactions involving the ribose 2′-OH groups of the RNA were observed and these interactions discriminate DNAs from being SDN1 substrates[16]. The SDN1 catalytic pocket seizes the RNA using two loops, the amino acid sequences of which are partially conserved in some DEDD members but highly diversified in others (Supplementary Fig. 1). Loop 1 is small and its backbone could be well superimposed to the corresponding loops in other DEDD members (Supplementary Fig. 2k). Nevertheless, RNA recognition by loop 1 requires the side chains of His223 and Ser224, suggesting the importance of amino acid composition in this loop region. In contrast, loop 2 is larger and more flexible, which can be partially anchored upon RNA binding (the part of the loop visible in the structure was mostly RNA-interacting region, Supplementary Fig. 2j). Structural superimposition between SDN1 and other DEDD members shows large differences in the size of loop 2 and the orientation of the proceeding α4 of SDN1 DEDDh (Supplementary Fig. 2k). Regardless of the sequence divergence and distinct structural characteristics of loop 2 among DEDD members, some DEDD exonucleases still use the corresponding loops in substrate recognition, e.g., in the structure of 3′ hExo1 in complex with a stem-loop RNA and a stem-loop binding protein (PDB code: 4L8R), Lys276 in the corresponding loop is engaged in RNA binding[41]. The structural observation of the two loops in SDN1 functioning in substrate recognition may be applicable to other DEDD exonucleases.

Our structural studies reveal that the SDN1 CTD assumes an RRM fold. Unlike many RRMs, which exhibit strong substrate binding affinities (in the nM range) and sequence specificities, SDN1 RRM binds RNA weakly in a sequence independent mode. Moreover, canonical RRMs usually fold into a βαββαβ topology, while our structural and NMR titration studies show that SDN1 RRM is composed of a six-stranded antiparallel β sheet with the unconventional extension of β4 and β5, and the two extended β strands are involved in RNA binding. Despite its low RNA binding affinity, the RRM domain is critical to the overall RNA binding affinity and processive enzyme activity of SDN1 by working cooperatively with the DEDDh domain—while the DEDDh domain attacks the 3′ end of an ssRNA, the RRM binds the 5′ region of the RNA.

In vitro, the SDN1 RRM domain is important for the activity of SDN1 on other types of substrates, such as RNA duplexes with single-stranded termini and AGO1-bound miRNAs. The activity of SDN1 on RNA duplexes with unpaired 3′ termini may be relevant in vivo. For example, stem-loop RNAs with 3′ overhangs such as pri-miRNAs could be in vivo targets of SDN1. The high complementary between miRNAs and target RNAs in plants[42]

may result in the dissociation of miRNA/target mRNA duplexes from AGO1, in analogy to miRNA release from hAgo2 induced by highly complementary target RNAs observed both in vitro and in cells[22,39]. Our in vitro data suggests that SDN1 may trim the miRNA 3′ ends in released miRNA/target mRNA duplexes with the RRM domain binding to the target mRNA strand to facilitate trimming.

The 3′ end trimming of an AGO1-bound miRNA requires the 3′ end to be released from the AGO1 PAZ domain. The observation of target RNA-triggered trimming and tailing of AGO1-bound miRNAs in Drosophila[10], along with structural support for target RNA displacing the 3′ end of the guide strand from AGO PAZ[21], implies that target RNAs promote the access to or activities on AGO1-bound miRNAs by the trimming or tailing enzymes. In our in vitro SDN1 assays on AGO1-bound miRNAs, we found that the RNA binding activity of the SDN1 RRM domain is critical. We propose a model on how target RNAs and SDN1 act in concert to enable the trimming of AGO1-bound miRNAs. In this model, the target RNA displaces the 3′ end of the miRNA from the AGO1 PAZ domain, enabling SDN1 to gain access to the miRNA 3′ end for trimming. Meanwhile, the RRM domain binds the target strand to enhance processivity of the enzyme, as illustrated in Fig. 6c.

In this study, we also revealed a direct interaction between SDN1 and the AGO1/10 PAZ domain. The phenomena of exonucleases interacting with Argonaute proteins to facilitate the truncation of AGO/PIWI-associated small RNAs have been observed elsewhere. Drosophila Nibbler was co-immunoprecipitated with AGO1 and interacts with PIWI to process pre-piRNA 3′ ends in ovaries;[30,32] Neurospora QIP interacts with QDE-2 (an AGO protein) to unwind and trim pre-milRNAs;[27] Dis3l2 binds hAgo2 to truncate some hAgo2-bound miRNA species in an RNA-dependent manner;[43] silkworm Trimmer indirectly interacts with PIWI through the Tudor protein Papi/Tdrkh[34]. Most of the exoribonucleases that act on AGO-associated small RNAs could interact with RISC directly or indirectly. The interaction between SDN1 and AGO1 PAZ may help to stabilize the reaction complex and enhance the trimming process.

## Methods

**Plasmid construction, protein expression, and purification.** Full-length or truncated Arabidopsis SDN1 cDNA was cloned into pET28-SMT3 (with a His-SUMO tag) for protein expression in E. coli BL21 (DE3) cells. The mutation sites of the SDN1 CTD mutant (E329A/E330A/E332A) were predicted by the UCLA MBI SERp Server to enhance protein crystallizability[44]. The point mutations were introduced into SDN1 by PCR with a pair of overlapping primers containing the mutations with the QuikChange Site-Directed Mutagenesis Kit (Stratagene). All other SDN1 mutants were constructed using this method. DNA fragments corresponding to AGO1 or AGO10 PAZ and MID-PIWI domains were PCR amplified from AtAGO1 or AtAGO10 constructs[45] and cloned into a pET28-MBP vector in which the MBP gene was inserted between NdeI and BamHI sites of pET28. All constructs were validated through DNA sequencing.

All SDN1 proteins and AGO proteins were expressed in E. coli BL21 (DE3). In general, cells were cultured at 37 °C to an $OD_{600}$ of 0.6, then isopropyl β-D-1-thiogalactopyranoside (IPTG) was added to a final concentration of 0.2 mM, after which cells were grown for about 16 h at 18 °C. Cells were collected via centrifugation, resuspended in Lysis Buffer (20 mM Tris, pH 8.0, 500 mM NaCl, 25 mM Imidazole, pH 8.0) and lysed via an ultrahigh-pressure homogenizer (JNBIO). Cellular debris were removed by centrifugation and the supernatant was applied to a HisTrap FF prepacked column (GE Healthcare). The target protein was eluted via AKTA purifier (GE Healthcare) using elution buffers with a gradient concentration of imidazole (20 mM Tris, pH 8.0, 500 mM NaCl, 25–500 mM imidazole, pH 8.0). Then ULP protease was used to remove the 6xHis-SUMO-tag, and dialysis was applied to remove the high salt (20 mM Tris, pH 8.0, 200 mM NaCl). Then the sample was loaded onto the HisTrap column again and this time the flow-through was collected. The protein sample was diluted to a final NaCl concentration of 100 mM and further purified through ion exchange chromatography with a Q FF column (GE Healthcare). The protein was eluted with a linear gradient of 0.1–1 M NaCl mixed with a buffer containing 20 mM Tris, pH 8.0, and 0.1% β-Mercaptoethanol via AKTA purifier. The eluted protein was concentrated and

further subjected to size exclusion chromatography with a Superdex 75 HiLoad 16/60 column (GE Healthcare) to achieve high homogeneity in the protein sample (GF Buffer: 10 mM Tris, pH 8.0, 100 mM NaCl, 1 mM dithiothreitol). The final purified SDN1 protein was analyzed on a 15% SDS-PAGE gel, and concentrated to approximately 40 mg/mL. Then the sample was aliquoted, flash frozen by liquid nitrogen and stored at −80 °C.

Seleno-methionine (Se-Met) or isotope $^{13}$C/$^{15}$N-labeled SDN1 proteins were expressed in BL21 (DE3) in M9 medium supplied with L-selenomethionine or with $^{15}$NH$_4$Cl and [$^{13}$C] glucose, respectively. Derivative SDN1 proteins were purified in the same way as native SDN1 proteins.

**Crystallization and data collection.** SDN1 protein was diluted in GF Buffer to 15 mg/mL and a 10 nt single-stranded RNA was added in a protein/RNA molar ratio of 1:1.2. The sample was incubated on ice for 30 min and screened for crystals using hanging drop vapor diffusion at 20 °C. The initial crystals were obtained in 10 mM MgCl$_2$, 50 mM MES, pH 5.6, 1.8 M Li$_2$SO$_4$, and optimized to larger crystals in 10 mM MgCl$_2$, 50 mM MES, pH 5.2, 1.45 M Li$_2$SO$_4$. Crystals of Se-Met-substituted SDN1-ssRNA complex were grown under the same condition. SDN1 CTD (residues 309–405, E329A/E330A/E332A) crystals were obtained by hanging drop vapor diffusion at 20 °C in 21% PEG 4 K, 0.2 M NH$_4$Cl, and crystals of Se-Met-substituted SDN1 CTD were grown in 23% PEG 3350, 0.2 M NH$_4$-citrate. All crystals were cryo-protected by the reservoir solution supplemented with 20% ethylene glycol (SDN1-ssRNA) or 20% glycerol (SDN1 CTD) before flash cooling in liquid nitrogen.

All diffraction data were collected at beamline BL-17U at the Shanghai Synchrotron Radiation Facility (SSRF). Data were processed and scaled using the HKL2000 package. Initial phases were determined by Single-wavelength Anomalous Diffraction (SAD) using the Se-Met-substituted SDN1 proteins or by Molecular Replacement (MR) using Se-Met SDN1 ΔC-ssRNA structure as search model.

**Structure determination and refinement.** The Se-Met SDN1 ΔC-ssRNA structure and the RRM structure were solved using SAD with the AutoSol and Auto-Build programs embedded in the PHENIX suite[46]. Some of the residues were manually built based on the electron density map using Coot[47]. The models were then refined against the diffraction data using the Refmac5 program of CCP4[48] and phenix.refine in PHENIX[46]. The diffraction data from a native SDN1-ssRNA crystal was collected and processed by MR using Se-Met SDN1 ΔC-ssRNA structure as the search model. The final model of the SDN1 ΔC-ssRNA was built based on the improved density map of the native dataset after refinement. During refinement, at least 5% of randomly selected data was set aside for free R-factor cross validation calculations. Water molecules were added either automatically or manually using Coot[47]. Sulfate, citrate, and metal ions were modeled in the refinement until the last few cycles. Structural figures were prepared with Pymol[49].

**NMR spectroscopy of SDN1 RRM and the RNA titration assay.** $^{13}$C/$^{15}$N-labeled SDN1 CTD (residues 305–405) was used in the standard NMR spectroscopy experiments to assign its $^1$H, $^{13}$C, and $^{15}$N backbone and sidechain chemical shifts, and to obtain the NOE-based restraints[50,51]. The spectra included the two-dimensional (2D) $^{13}$C-edited Heteronuclear Single Quantum Coherence (HSQC) and $^{15}$N-edited HSQC; the three-dimensional (3D) HNCO, HNCACO, HNCA, HNCACB, CBCA(CO)NH, HBHA(CO)NH, $^{15}$N-resolved HSQC-TOCSY, HCCH-TOCSY for both aliphatic and aromatic regions, $^{15}$N-resolved HSQC-NOESY and $^{13}$C-resolved HSQC-NOESY in both aliphatic and aromatic resonances[51,52]. All NMR experiments were performed at 297 K using a Varian Unity Inova 600 MHz NMR spectrometer, with all spectra processed with NMRPipe[53] and analyzed using Sparky[54].

The RNA (5′-AGCCCAUUAG-3′) was dissolved in 50 mM Na$_2$HPO$_4$, pH7.4, 100 mM NaCl, 10% D$_2$O; and the pH was adjusted to pH 7.4 by addition of NaOH. Then the RNA was gradually added to the $^{15}$N-labeled SDN1 C-terminus (309–405) to a protein/RNA ratio of 1:0, 1:0.5, 1:0.7, 1:1, 1:1.2, 1:1.5, 1:1.7, and 1:2. The $^1$H-$^{15}$N HSQC spectrum was recorded every time when a portion of the RNA was added and the weighted average chemical shift perturbations ($\Delta\delta_{avg}$) of each backbone amide resonance were calculated using the equation: $\Delta\delta_{avg} = [\Delta\delta_H^2 + (0.2 \times \Delta\delta_N)^2]^{1/2}$, where $\Delta\delta_H$ and $\Delta\delta_N$ are the differences in the $^1$H and $^{15}$N chemical shifts, respectively. Amino acids with chemical shift changes greater than 0.05 p.p.m. are considered as ones showing large changes.

**In vitro enzymatic activity assays.** All RNA oligonucleotides were synthesized at Dharmacon or IDT, and labeled by $^{32}$P at their 5′ ends. Labeled miR166 was annealed to its cold target RNA in a 1:1.5 molar ratio, and the RNA duplexes were analyzed on a native gel to ensure that all miR166 were in duplexed form. 50 ng/μL SDN1 wild-type or mutant proteins were incubated with 2 nM labeled RNA substrates at room temperature for 30 min in 50 mM Tris, pH 8.0, 135 mM KCl, 2.5 mM MgCl$_2$ (or 2.5 mM MnCl$_2$ as indicated), 1 mM DTT, 40 units RNasein (Promega). Reactions were stopped by adding deionized formamide followed by boiling at 95 °C for 10 min. The RNAs were then resolved on 15% denaturing polyacrylamide gels.

**Binding assays.** Wild-type SDN1 or mutants (0.5 μM–0.5 mM) were incubated with 2 nM $^{32}$P-labeled RNAs on ice for 30 min in 50 mM Tris, pH 8.0, 100 mM NaCl, 1 mM DTT in a total volume of 10 μL. The samples were analyzed on 8% native PAGE gels with cold 0.5× TBE buffer.

**UV crosslinking assays.** 5′-iodoU modified miR166 target RNAs were ordered from Dharmacon, 5′ end $^{32}$P-labeled, and gel purified before annealing to miR166 in 1:1.5 molar ratio. Ten nanomolar RNA was incubated with 5 μM wild-type or mutant SDN1 proteins in 50 mM Tris, pH 8.0, 100 mM NaCl, 1 mM DTT in a total volume of 10 μL at room temperature for 30 min. The mixture was subjected to UV irradiation for 30 min on ice. The UV light was filtered through a polystyrene Petri dish to retain UV light at the wavelength of approximately 312 nm. For the SDN1$^{TEV}$ mutant, crosslinking products were digested with 5 units of TEV protease (ProTEV Plus, Promega) on ice for 1 h. All samples were incubated with SDS loading dye at 95 °C for 10 min before being resolved on 12% or 15% SDS-PAGE gels.

**RNA competition assays.** The 5′ end $^{32}$P-labeled miR166 (approximately 25 fmoles for each reaction) was first incubated with 5 pmoles of wild-type or mutant SDN1 proteins for 20 min in buffer containing 20 mM Tris-HCl, pH 8.0, 135 mM KCl, 1 mM DTT, 0.5 mM EDTA, pH 8.0, and 5% Glycerol. This buffer allows substrate binding but does not support catalytic reactions due to the lack of Mg$^{2+}$. Then MgCl$_2$, or a mixture of MgCl$_2$ and 0.1 nmoles of cold miR166, was added to initiate exonucleolytic reactions. The final concentration of Mg$^{2+}$ was 5 mM, and the final molar ratio of labeled miRNA: enyzme: cold RNA competitor was 1:200:4000.

**Generation of anti-SDN1 antibodies.** SDN1 polyclonal antibodies were generated by immunizing a rabbit with the full-length SDN1 recombinant protein. The antibodies were affinity-purified using SDN1 protein expressed from *E. coli* as described below. First, 1 mL medium was prepared by swelling and washing 0.2857 g freeze-dried, CNBr-activated Sepharose 4B powder with 60 ml HCl (1 mM) for 15 min. 5–10 mg SDN1 recombinant protein was dialyzed against coupling buffer (0.1 M NaHCO$_3$, pH 8.3, 0.5 M NaCl), concentrated to 1.5 ml (3.3–6.7 mg/ml) and mixed with the medium. The mixture was incubated on an end-over-end mixer for 1 h at room temperature or overnight at 4 °C. The medium was packed into a column and excess SDN1 protein was washed away with at least 5 column volumes (cv) of coupling buffer. The medium was transferred to 0.1 M Tris-HCl buffer, pH 8.0 or 1 M ethanolamine, pH 8.0 for 2 h to quench any remaining active groups. The medium was then washed with two buffers for at least three rounds. In each round, the medium was first washed with 0.1 M NaAc, pH 4.0, 0.5 M NaCl, followed by another wash with 0.1 M Tris-HCl, pH 8.0, 0.5 M NaCl. Then the medium was washed with TBS (0.15 M NaCl, 20 mM Tris-HCl, pH 7.4). Rabbit serum was diluted 1:2 with TBS, passed through a 0.2 μm filter and incubated with the medium on an end-over-end mixer overnight at 4 °C. The medium was then washed with 5 cv of TBS, 10 cv of 0.5 M NaCl, 20 mM Tris-HCl, pH 7.4, 0.2% Triton- X-100, and 5 cv of TBS. The antibody was eluted in fractions with 0.15 M NaCl, 0.2 M Glycine-HCl, pH 2.0, and 10% (v/v) of 2 M Tris-HCl, pH 8.5, was added to each fraction. The fractions containing anti-SDN1 antibodies were determined by dot blotting followed by protein staining. The fractions containing SDN1 antibodies were pooled, aliquoted and stored at −80 °C.

To demonstrate that the SDN1 antibodies recognize SDN1, total proteins from wild-type and *rdr6 sdn1 sdn2* 12-day-seedlings were extracted, and western blotting was performed using 1 μg/ml SDN1 antibody (1: 2000 dilution).

**In vitro pull-down assays.** Approximately 40 μg of purified MBP-tagged, truncated AGOs and MBP alone were mixed with 1 μg of purified wild-type or mutant SDN1 proteins in 20 mM Tris-HCl, pH 8.0, 100 mM NaCl, 0.1 mM MgCl$_2$, 1 mM DTT, 5% Glycerol and 0.2 mg/ml BSA. Then 20 μL pre-equilibrated dextrin beads (GE Healthcare Dextrin High Performance) were added to each mixture and the samples were incubated at 4 °C overnight on a shaker. All samples were then centrifuged at 86×*g* for 30 s to collect beads. The beads were washed ten times using wash buffer (20 mM Tris-HCl, pH 8.0, 100 mM NaCl, 0.5% NP-40), and SDS loading dye was added to the beads. After being heated at 95 °C for 10 min, the samples were resolved on 12% SDS-PAGE gels followed by western blotting experiments with 1:1000 diluted anti-SDN1 antibody and 1:1000 diluted HRP-labeled goat anti-rabbit IgG (Beyotime).

**In vitro enzymatic assays on AGO1-bound miRNAs.** Immunoprecipitation (IP) of AGO1 was performed as described[13]. In brief, 1 g of 12-day-old wild-type seedlings was ground to a fine powder in liquid nitrogen and resuspended in 1.5 ml IP buffer. The extract was incubated with 4 μl anti-AGO1 antibody (Agrisera). Then the protein-protein complexes were captured by 30 μL pre-cleared Dynabeads-Protein-A (Life Technologies). A portion of the IPs was subjected to western blotting to detect AGO1. The rest of the AGO1 IPs was resuspended in reaction buffer (50 mM Tris-HCl, pH 8.0, 150 mM NaCl, 1 mM DTT, 2.5 mM MnCl$_2$, 1 mM ATP), and aliquoted to three equal portions, to which wild-type SDN1, SDN1$^{D283A}$, and SDN1$^{2FA}$ were each added. Each reaction contained 275 pmoles of SDN1 proteins and less than 5 pmoles of small RNAs present in AGO1 IPs. After incubation at room temperature for 2 h, the beads were collected for RNA extraction and small RNA library

construction. The RNA extraction, small RNA library construction and bioinformatics analysis were performed as described[15].

## Data availability

Structural coordinates and diffraction structural files have been deposited in the Protein Data Bank under the PDB codes of 5Z9X (SDN1 ∆C-RNA complex) and 5Z9Z (SDN1 CTD). Data is available from the authors upon reasonable request.

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

## Acknowledgements

We thank Dr. Xiuren Zhang for MBP-tagged AGO1/10 expression plasmids, Drs. Hong Ma and Jun Wang for sharing instruments, and the staff of beamline BL17U of Shanghai Synchrotron Radiation Facility where the X-ray diffraction data were collected. This work was supported by the National Natural Science Foundation of China (31230041 to

J.M. and 91640102 to Y.H.) and the National Basic Research Program of China (2011CB966304 and 2012CB910502) to J.M. and Y.H., the Young 1000 Tanlent Project and Shuguang Project to J.M., the Strategic Priority Research Program of the Chinese Academy of Sciences (XDB08010202) to Y.H., and National Institutes of Health GM061146 to X.C.

## Author contributions

Y.H. initiated the project, purified proteins, obtained SDN1-ssRNA crystals, and solved the structure, J.C. purified proteins, obtained C-terminal domain crystals, performed in vitro enzymatic assays, binding assays, UV crosslinking assays and pull-down assays. L.L. performed in vitro enzymatic assays on AGO1-bound miRNAs and C.Y. processed the small RNA sequencing data. J.Gu performed the NMR titration assay and analyzed the data at C.C.'s lab. Y.H., J.M., and J.C. collected X-ray diffraction data. Y.H. and J.M. built the initial models. Y.H., J.M., W.R., and J.C. refined the structures. J.C. and X.C. wrote the original draft. X.C., J.M., Y.H., J.Ga., J.C., and L.L. reviewed and revised the manuscript. J.M. and X.C. conceived the project and supervised research.Data availabilityStructural coordinates and diffraction structural files have been deposited in the Protein Data Bank under the PDB codes of 5Z9X (SDN1 ΔC-RNA complex) and 5Z9Z (SDN1 CTD). Data is available from the authors upon reasonable request.

## Additional information

**Competing interests:** The authors declare no competing interests.

