## [Peer Review File · Nature Communications]

Reviewers' Comments:

Reviewer #1:

Remarks to the Author:

miRNAs, a big family of small-noncoding RNAs, play critical roles in various biological processes. The amount of miRNAs is determined by miRNA production and degradation. Years ago, the Chen group has found that Small RNA degrading nucleases (SDNs), a family of 3' to 5' exonucleases, can degrade short RNAs in vitro and limit the accumulation of miRNAs in vivo. Latest study from the same group showed that SDN1 acts on the AGO1-bound miRNAs to cause their 3' truncation. In this setting, the team took the torch to further study biochemical mechanism of how SDN1 degrades small RNAs. Here, Chen et al reported the crystal structures of SDN1 (N-terminal and DEDDh domain) with 9-nt small RNAs. The authors also solved the crystal structure of the C-terminal domain and found that it adopts an RNA Recognition Motif (RBM) fold. By a series of elegant assays, they tested how SDN1 bound to and degraded various forms of RNAs including single-stranded RNA, small RNA-targeted RNA duplex (with 5' region of miRNA unpaired). They found that the C-terminal RRM domain binds to the 5' regions of RNA substrates or target stranded in the miRNA-target RNA duplexes and such conformation could promote DEDDh domain to attack the 3'ends of miRNAs for trimming. Finally, the authors also showed that SDN1 could directly interact with the PAZ domains of AGO proteins; thus facilitating its access to AGO1-bound miRNAs and trim their 3'ends.

Overall speaking, this is an excellent work that provides new insight into the mechanism of how SDN1 acts on miRNA substrates in different scenarios. The study also shed light on a general 3' end trimming mechanism of 3' to 5' riboexonucleases in the metabolism of various species of RNAs. The experimental designs are clever and data are mostly solid. The study clearly pushes the field forward and also has general interest in different fields. I only have some minor comments for possible polishing of ms, otherwise, the ms is in a good shape for publication in NC.

- 1) Figure 1 A and B—make the colors of DEDDh consistent in two panels.
- 2) Figure 3D and J; EMSA shows a typical sigmoidal curve of protein-RNA binding—It looks like that SDN1 has two domains that cooperatively bind to RNA (here N-terminal domain and RBM at the C-terminal). If this were the case, would it be more accurate to label it as apparent K_d and discuss it?
- 3) The discussion part, the authors mentioned the possible substrates might be pri-miRNAs with some overhanging. If the authors mine miRNA databases from sdn mutant, could one see any inaccuracy of 5'ends of miRNAs? Please discuss it in the discussion part.

Reviewer #2:

Remarks to the Author:

In this paper, Chen and colleagues reported the crystal structure of Arabidopsis SDN1 in complex with a single-stranded RNA. The structure showed the DEDDh domain interacting with 3' end 4 nucleotides of the bound RNA while the C-terminal domain is completely disordered. However, their biological assay indicated that the C-terminal domain is essential to the sufficient RNA-binding of SDN1 in a cooperative manner. They also reported that the SDN1 interacts with the PAZ domain of AGO1 and AGO10 even in the absence of RNA.

Overall, this paper provides the structural basis for the 3' end trimming of small RNAs by SDN1. But the authors need to consider the following points.

Major points:

1. (lines 27-30) The sentence of 'the crystal structure of Arabidopsis SDN1 without its C-terminal domain' is confusing because the authors reported in the main text 'Full-length, recombinant Arabidopsis SDN1 and a 10 nt ssRNA were used in crystallization in lines 110-111. Being the

abstract, this wording should be clarified so the reader can enter the main body of the paper with a clear understanding of what the authors intend to present. The crystal packing should be shown in Supplementary information so that there is enough space for the disordered C-terminal domains. Also, the authors may want to test if the C-terminal domain got cleaved during crystallization.

2. Although proposing a model of forming a quaternary complex of AGO1, SDN1, small RNA and target RNA, the authors showed only the interaction between SDN1 and small RNA. Trimming assays in Fig. 5d do not support that SDN1's trimming happens on AGO1. The authors need to demonstrate the existence of the quaternary complex experimentally.

3. Can the authors exclude a possibility that the RRM domain of two SDN1 actually dimerizes and the remainder (i.e., two of the NTD and DDDh) pinches the small RNA as seen in the crystal structure (Supple Fig. 1b)? SEC analyses will answer the question.

4. (Fig. 2a-2d) The simulated annealing Fo-Fc maps of the protein part should be shown because the adjacent loop2 was completely disordered.

5. The authors concluded that the RRM domain binds to the middle part and 5' side of the small RNA based on the result that SDN1 deltaC didn't bind to the small RNA. That said, the authors did not show the direct evidence that the RRM physically interacted with the 5-iodoU. The experiments are required to show a physical interaction between the RRM and the 5-iodoU.

6. (line 187) Why were the three glutamate residues mutated?

7. The mutants used in Fig. 2f need to be tested for stability.

8. No large chemical shift was observed for F356, and thus the author concluded that F356 might not be involved in RNA interaction on page 8. Meanwhile, the authors mutated F356 on page 9, and the mutant showed a much lower affinity for miRNA (Supple Fig. 3c-d). The authors need to explain this discrepancy experimentally.

9. (Fig. 2d) How did the author interpret that the density was derived from a magnesium ion but not from water? 1.6 sigma is quite low to put a magnesium ion into the density. The authors should show a simulated annealing Fo-Fc omit map at 4-5 sigma.

10. (Fig. 4f-g) The band intensities are very weak. Since the model (Fig. 4e, actually one of the main conclusions in this paper) heavily relies on these results, the authors should repeat the experiments to provide the strong evidence.

11. (lines 347-348) The authors excuse that an activity of a single nucleotide trimming is due to contamination of nuclease in SDN1 D283A. If so, this uncertainty is worrisome with respect to the results of the rest of the experiment.

12. Is there any possibility that the observed RNA recognition is due to the crystal packing? To validate the concern, the authors need to test SDN1 mutations of Q21 and S22.

Minor points

13. The sequences of RNAs used in the EMSA (Fig. 3) should be provided.

14. (line 104) *in vitro* should be italic.

15. (Supple Fig. 1b) the authors should show a simulated annealing Fo-Fc omit map.

16. (Fig. 1c) It is interesting that the 3' end 4 nucleotides are recognized by the negatively charged region. Can the authors explain why?
17. (line 249) Supple Fig. 2f may be 2e.
18. (line 216) Is Glu382 correct because it is depicted as G382 in Fig. 3g?
19. (Fig. 4 d) what are the other bands without asterisks?
20. The western blot data need to be shown for any immunopurified samples.

We sincerely thank both reviewers for reading our manuscript with great care, and for their helpful comments, encouragement, and criticism. Based on the comments and suggestions, we have carefully revised our manuscript with major changes highlighted in the manuscript text. The following are our point-by-point responses to the reviewers' comments.

Reviewer #1 (Remarks to the Author):

miRNAs, a big family of small-noncoding RNAs, play critical roles in various biological processes. The amount of miRNAs is determined by miRNA production and degradation. Years ago, the Chen group has found that Small RNA degrading nucleases (SDNs), a family of 3' to 5' exonucleases, can degrade short RNAs in vitro and limit the accumulation of miRNAs in vivo. Latest study from the same group showed that SDN1 acts on the AGO1-bound miRNAs to cause their 3' truncation. In this setting, the team took the torch to further study biochemical mechanism of how SDN1 degrades small RNAs. Here, Chen et al reported the crystal structures of SDN1 (N-terminal and DEDDh domain) with 9-nt small RNAs. The authors also solved the crystal structure of the C-terminal domain and found that it adopts an RNA Recognition Motif (RBM) fold. By a series of elegant assays, they tested how SDN1 bound to and degraded various forms of RNAs including single-stranded RNA, small RNA-targeted RNA duplex (with 5' region of miRNA unpaired). They found that the C-terminal RRM domain binds to the 5' regions of RNA substrates or target stranded in the miRNA-target RNA duplexes and such conformation could promote DEDDh domain to attack the 3'ends of miRNAs for trimming. Finally, the authors also showed that SDN1 could directly interact with the PAZ domains of AGO proteins; thus facilitating its access to AGO1-bound miRNAs and trim their 3'ends.

Overall speaking, this is an excellent work that provides new insight into the mechanism of how SDN1 acts on miRNA substrates in different scenarios. The study also shed light on a general 3' end trimming mechanism of 3' to 5' riboexonucleases in the metabolism of various species of RNAs. The experimental designs are clever and data are mostly

solid. The study clearly pushes the field forward and also has general interest in different fields. I only have some minor comments for possible polishing of ms, otherwise, the ms is in a good shape for publication in NC.

1) Figure 1 A and B—make the colors of DEDDh consistent in two panels.

We thank the reviewer for carefully inspecting our figures. Figure 1 was edited as suggested.

2) Figure 3D and J; EMSA shows a typical sigmoidal curve of protein-RNA binding—It looks like that SDN1 has two domains that cooperatively bind to RNA (here N-terminal domain and RBM at the C-terminal). If this were the case, would it be more accurate to label it as apparent K_d and discuss it?

We thank the reviewer for the constructive suggestion. The sigmoidal curve from EMSA suggests that the DEDDh and RRM domains work cooperatively in substrate binding. We have replaced ' K_d ' with ' $K_{d, app}$ ' and discussed it in the manuscript, shown as highlighted texts.

3) The discussion part, the authors mentioned the possible substrates might be pri-miRNAs with some overhanging. If the authors mine miRNA databases from sdn mutant, could one see any inaccuracy of 5'ends of miRNAs? Please discuss it in the discussion part.

We mentioned this possibility because we found that SDN1 can act on duplexed RNAs with 3' single-stranded overhangs of 4 nt or longer in vitro and pri-miRNAs have a hairpin structure with 5' and 3' single-stranded overhangs. We do not have evidence that SDN1 acts on pri-miRNAs in vivo. If it acts on pri-miRNAs in vivo to trim the 3' single-stranded regions, it is not expected to affect the precision of pri-miRNA processing by DCL1, as processing occurs at 15-17nt from a bulge in the stem region or at a certain distance from the loop (these regions are not expected to be affected by SDN1).

References:

1. Zhu, H. et al. Bidirectional processing of pri-miRNAs with branched terminal loops by Arabidopsis Dicer-like1. *Nat Struct Mol Biol* 20(9):1106-15 (2013)
2. Cuperus, J.T. et al. Identification of MIR390a precursor processing-defective mutants in Arabidopsis by direct genome sequencing. *Proc Natl Acad Sci USA* 107, 466-471 (2010)
3. Mateos, J.L. et al. Identification of microRNA processing determinants by random mutagenesis of Arabidopsis MIR172a precursor. *Curr Biol* 20, 49-54 (2010)
4. Song, L., Axtell, M. & Fedoroff, N. RNA secondary structural determinants of miRNA precursor processing in Arabidopsis. *Curr Biol* 20, 37-41 (2010)
5. Werner, S. et al. Structure determinants for accurate processing of miR172a in Arabidopsis thaliana. *Curr Biol* 20, 42-48 (2010)
6. Addo-Quaye, C. et al. Sliced microRNA targets and precise loop-first processing of MIR319 hairpins revealed by analysis of the *Physcomitrella patens* degradome. *RNA* 15, 2112–2121 (2009)
7. Bologna, N.G. et al. A loop to base processing mechanism underlies the biogenesis of plant microRNAs miR319 and miR159. *EMBO J* 28, 3646–3656 (2009)

Reviewer #2 (Remarks to the Author):

In this paper, Chen and colleagues reported the crystal structure of Arabidopsis SDN1 in complex with a single-stranded RNA. The structure showed the DEDDh domain interacting with 3' end 4 nucleotides of the bound RNA while the C-terminal domain is completely disordered. However, their biological assay indicated that the C-terminal domain is essential to the sufficient RNA-binding of SDN1 in a cooperative manner. They also reported that the SDN1 interacts with the PAZ domain of AGO1 and AGO10 even in the absence of RNA.

Overall, this paper provides the structural basis for the 3' end trimming of small RNAs by SDN1. But the authors need to consider the following points.

Major points:

1. (lines 27-30) The sentence of 'the crystal structure of Arabidopsis SDN1 without its C-terminal domain' is confusing because the authors reported in the main text 'Full-length, recombinant Arabidopsis SDN1 and a 10 nt ssRNA were used in crystallization in lines 110-111. Being the abstract, this wording should be clarified so the reader can enter the main body of the paper with a clear understanding of what the authors intend to present. The crystal packing should be shown in Supplementary information so that there is enough space for the disordered C-terminal domains. Also, the authors may want to test if the C-terminal domain got cleaved during crystallization.

We thank the reviewer for pointing out the ambiguous phrasing. We rephrased this sentence into 'the crystal structure of Arabidopsis SDN1 (residue 2-300)' in Abstract, shown as highlighted texts. We also added an SDS-PAGE gel analysis of the SDN1-ssRNA crystals and a depiction of crystal packing as Supplementary Fig. 2d,e. These data suggest that SDN1 CTD is not cleaved during crystallization and there is enough space in the crystal lattice to hold the CTD. Given the good qualities of the CTD and the full-length SDN1 proteins in purification, it is likely that the RRM domain is not in a disordered conformation in the SDN1-ssRNA crystal, but rather the linker between the DEDDh and RRM domains is flexible, and the RRM domain adopts different orientations in the crystal lattice. Therefore, the RRM domain is missing in the electron density, even though full-length SDN1 is in the crystal.

2. Although proposing a model of forming a quaternary complex of AGO1, SDN1, small RNA and target RNA, the authors showed only the interaction between SDN1 and small RNA. Trimming assays in Fig. 5d do not support that SDN1's trimming happens on AGO1. The authors need to demonstrate the existence of the quaternary complex experimentally.

First, we believe that trimming assay in Fig.5d do suggested that SDN1 can trim the 3' ends of AGO1-bound miRNAs. The *in vitro* trimming assays in Fig. 5d-g were conducted with AGO1 immunoprecipitates from *Arabidopsis*, i.e., the miRNAs used for sequencing in this assay were AGO1-bound endogenous miRNAs.

Second, we agree that the quaternary complex in Fig. 5b is purely our model of how SDN1 works on AGO1-bound miRNAs, and currently we cannot demonstrate its existence experimentally, since the interactions between SDN1, AGO1-miRNAs and target RNAs could be transient. But, we showed that SDN1 interacts with the PAZ domain of AGO1/10 by *in vitro* pull-down assay, and SDN1 trims free miRNA/target RNA duplexes with the RRM domain binding to the 5' region of the target RNA. More importantly, in Fig. 5d-g, we showed that the RRM mutant 2FA exhibited reduced enzymatic activity on AGO1-bound miRNAs comparing to wild-type SDN1. The impact of mutations in SDN1 RRM domain suggested that it is possible that the AGO1 immunoprecipitates contain target RNAs, as seen previously ¹. In addition, previous work demonstrated that artificial target RNAs, the Shot Tandem Target Mimic RNAs (STTMs), can reduce the levels of cognate miRNAs or siRNAs in *Arabidopsis* depending on SDNs ². Based on these experimental evidence, we propose a model that SDN1 trims the 3' end of AGO1-bound miRNA in the presence of target RNA.

Reference:

1. Carbonell, A. et al. Functional analysis of three *Arabidopsis* ARGONAUTES using slicer-defective mutants. *Plant Cell* 24(9):3613-29 (2012)
2. Yan, J. et al. Effective Small RNA Destruction by the Expression of a Short Tandem Target Mimic in *Arabidopsis*. *Plant Cell* 24(2):415-427 (2012).

3. Can the authors exclude a possibility that the RRM domain of two SDN1 actually dimerizes and the remainder (i.e., two of the NTD and DDDh) pinches the small RNA as seen in the crystal structure (Supple Fig. 1b)? SEC analyses will answer the question.

We thank the reviewer for this good point and the suggested test. To exclude the possibility of dimerization of the RRM domain, we conducted size exclusion chromatography (SEC) analysis using a GE Healthcare Superdex™ 200 Increase 10/300 GL prepack column. The results show that the retention volume of SDN1-ssRNA used in crystallization is almost the same as that of the SDN1 protein alone (shown in the following figure). When comparing with the retention volumes of standard proteins, ovalbumin (M_r 44kDa) is the closest one with a retention volume of around 14.2 mL. Therefore, it is reasonable to draw the conclusion that SDN1 remains as a monomer during crystallization.

4. (Fig. 2a-2d) The simulated annealing Fo-Fc maps of the protein part should be shown because the adjacent loop2 was completely disordered.

We thank the reviewer for the suggestion. We included the simulated annealing Fo-Fc map of loop2 (residues 253-259) as Supplementary Fig. 2j.

5. The authors concluded that the RRM domain binds to the middle part and 5' side of the small RNA based on the result that SDN1 deltaC didn't bind to the small RNA. That

said, the authors did not show the direct evidence that the RRM physically interacted with the 5-iodoU. The experiments are required to show a physical interaction between the RRM and the 5-iodoU.

We are sorry that the description in our previous manuscript may not have clearly explained the design and results of this experiment. We showed that SDN1 Δ C (NTD+DEDDh) did not crosslink to 5'-IUU RNA while the full-length SDN1 (NTD+DEDDh+RRM) did. This result ruled out the possibility that the NTD or DEDDh domains in full-length SDN1 crosslinked to the 5'-IUU RNA, leaving the only explanation that the RRM domain was crosslinked to the 5-iodoU sites located at the 5' region of the RNA. We edited the manuscript and Fig. 3m to better illustrate our points.

6. (line 187) Why were the three glutamate residues mutated?

The reason we used the RRM mutant (E329A/E330A/E332A) is that no crystals were obtained when wild-type SDN1 RRM proteins were initially used in crystallization. We then use the UCLA MBI Surface Entropy Reduction prediction (SERp) online server to identify potential sites for mutagenesis to reduce the protein's surface entropy and enhance crystallizability (the reason was mentioned in the Methods section, shown as highlighted texts). The E329A/E330A/E332A mutant is the top candidate given by the SERp server, and we were able to get crystals from this mutant.

7. The mutants used in Fig. 2f need to be tested for stability.

The SDS-PAGE gel analysis of all mutants used in Fig. 2f is shown in Supplementary Fig. 2i.

8. No large chemical shift was observed for F356, and thus the author concluded that F356 might not be involved in RNA interaction on page 8. Meanwhile, the authors mutated F356 on page 9, and the mutant showed a much lower affinity for miRNA (Supple Fig. 3c-d). The authors need to explain this discrepancy experimentally.

Since the two phenylalanines are conserved among RRM s and important for RNA binding based on studies of other RRM s, we designed the 2FA mutant containing double-site mutations of F318A and F356A in our EMSA and enzymatic assays; this was done before the NMR titration experiment. Unexpectedly, the NMR titration assay showed that F356 might not take part in RNA recognition. To confirm that mutagenesis of F356 did not impair the enzyme's RNA binding affinity nor catalytic activity, we made F356A and F318A mutants and conducted EMSA and enzymatic assays. The results showed that F356A exhibited similar binding and catalytic activity comparing wild type SDN1, while F318A and 2FA exhibited reduced RNA binding affinity and catalytic activity (Supplementary Fig. 4a-d). Therefore, both 2FA and F318A mutants have similar activities as RRM mutants, and 2FA mutant was used in the manuscript.

9. (Fig. 2d) How did the author interpret that the density was derived from a magnesium ion but not from water? 1.6 sigma is quite low to put a magnesium ion into the density. The authors should show a simulated annealing Fo-Fc omit map at 4-5 sigma.

We replaced Fig. 2d with a simulated annealing Fo-Fc omit map at 5.0 σ , which showed a magnesium ion in the active site. We actually tried to place a water molecule at the density site at first, but there were extra positive densities at the site after refinement. Moreover, the location of the density indicates that the corresponding atom coordinates oxygen atoms from D144, D228 and the 3' end scissile phosphate group of the RNA (Fig. 2d). These features are unlikely from a water molecule. Instead, the geometry of the coordination sphere agrees with the classic coordination pattern of the catalytic magnesium ion which is widely observed at the active sites of many DEDD exonucleases. Besides, the crystallization reservoir of the SDN1-ssRNA crystals contains 10 mM MgCl₂.

10. (Fig. 4f-g) The band intensities are very weak. Since the model (Fig. 4e, actually one of the main conclusions in this paper) heavily relies on these results, the authors should repeat the experiments to provide the strong evidence.

We agree that the band intensities in Fig. 4f-g are very weak. But the UV crosslinking assays in Fig. 4g were repeated for at least three times (shown in the following figure, in which Replicate 1 is the one in Fig. 4g), which suggested that the results are reproducible (red, dark green and yellow asterisks indicate full-length SDN1, SDN1 RRM and SDN1 Δ C crosslinked to RNA, respectively). The reason why the bands were weak should be related to the low efficiency of UV crosslinking, and the crosslinking efficiency might be affected by the low binding affinities of SDN1 proteins with the 5-iodoU RNAs ($K_{d, app}$ of SDN1 with a miRNA is in the μ M range).

11. (lines 347-348) The authors excuse that an activity of a single nucleotide trimming is due to contamination of nuclease in SDN1 D283A. If so, this uncertainty is worrisome with respect to the results of the rest of the experiment.

The explanation is based on our previous work (Yu, Y. et al, Plos Biol 2017), in which SDN1 D283A was included in the SDN1 enzymatic assay using AGO1/10 immunoprecipitates, and the results showed that the catalytic mutant D283A did not trim the AGO1/10-bound miRNAs. The procedure of the assay in our manuscript is identical to the one in the Yu, Y. et al, Plos Biol 2017 paper. Therefore, we believed that the single-nucleotide trimming is unlikely caused by SDN1 D283A.

12. Is there any possibility that the observed RNA recognition is due to the crystal packing? To validate the concern, the authors need to test SDN1 mutations of Q21 and S22.

To exclude the possibility, we made SDN1 mutations of Q21 and S22 that involved in crystal packing (or RNA interaction) and conducted EMSA. Mutagenesis of Q21 or S22 to alanine does not impair the enzyme's RNA binding affinity, as shown by EMSA assays in Supplementary Fig. 2b. Thus, Q21 and S22 should not take part in enzyme-substrate recognition, and the interactions between the 5' end of the RNA and Q21/S22 observed in the SDN1-ssRNA structure should result from crystal packing.

Minor points

13. The sequences of RNAs used in the EMSA (Fig. 3) should be provided.

The sequences of all RNAs used were provided in Supplementary Table 1.

14. (line 104) *in vitro* should be italic.

We thank the reviewer for carefully going through our manuscript. This mistake was corrected.

15. (Supple Fig. 1b) the authors should show a simulated annealing Fo-Fc omit map.

The panel was replaced by a simulated annealing Fo-Fc map of the RNA as Supplementary Fig. 2a.

16. (Fig. 1c) It is interesting that the 3' end 4 nucleotides are recognized by the negatively charged region. Can the authors explain why?

The negatively charged surface of the catalytic pocket is mostly contributed by the four invariant acidic residues D144/E146/D228/D283 and this property is conserved among other DEDD family members. The four acidic residues are critical for coordinating two divalent metal ions, which bind the 3' end scissile phosphate of the RNA and serve as Lewis acid during hydrolysis. Nevertheless, the catalytic pocket still contains other positively charged residues that are involved in RNA interaction, e.g., R185, R256 and H223 (shown in Fig. 2a-c).

17. (line 249) Supple Fig. 2f may be 2e.

We apologize for the mistake and thank the reviewer for reading our manuscript with great care. The figures were reorganized and renumbered to better accommodate the changes.

18. (line 216) Is Glu382 correct because it is depicted as G382 in Fig. 3g?

We thank the reviewer for carefully reading our manuscript and for the correction. We have corrected this mistake in the manuscript, shown as highlighted texts.

19. (Fig. 4 d) what are the other bands without asterisks?

In our crosslinking experiments using double-stranded RNAs, a pair of bands usually appeared together. We deduced from the sizes that the lower ones are crosslinking products of proteins with single-stranded RNAs, and the upper bands are crosslinking products of proteins with double-stranded RNAs that were not denatured into single strands during SDS-PAGE electrophoresis. We added an explanation for the bands in the legend of Fig. 4d, shown as highlighted texts.

20. The western blot data need to be shown for any immunopurified samples.

The MBP-tagged bait proteins and the SDN1 target protein used in the pull-down assays were purified from *E. coli*. All inputs used in the western blot were analyzed by SDS-PAGE shown in Supplementary Fig. 6a.

Reviewers' Comments:

Reviewer #1:

Remarks to the Author:

All of my concerns have been addressed well. Congratulations to the authors for this nice work.

Reviewer #2:

Remarks to the Author:

The Authors have addressed my concerns and I am happy with the revised manuscript without any further concerns.

RESPONSE TO REVIEWERS' COMMENTS:

We sincerely thank both reviewers for reading our manuscript with great care, and for their helpful comments, encouragement, and criticism. Based on the comments and suggestions, we have carefully revised our manuscript with major changes highlighted in the manuscript text. The following are our point-by-point responses to the reviewers' comments.

REVIEWERS' COMMENTS:

Reviewer #1 (Remarks to the Author):

All of my concerns have been addressed well. Congratulations to the authors for this nice work.

We thank the reviewer for the encouraging comment.

Reviewer #2 (Remarks to the Author):

The Authors have addressed my concerns and I am happy with the revised manuscript without any further concerns.

We thank the reviewer for the encouraging comment.